# Activation of the IRE1 RNase through remodeling of the kinase front pocket by ATP-competitive ligands

Elena Ferri[1,2], Adrien Le Thomas[3], Heidi Ackerly Wallweber[1], Eric S. Day[4], Benjamin T. Walters [5], Susan E. Kaufman[5], Marie-Gabrielle Braun[2], Kevin R. Clark[5], Maureen H. Beresini[5], Kyle Mortara[6], Yung-Chia A. Chen[3], Breanna Canter[2], Wilson Phung[7], Peter S. Liu[7], Alfred Lammens[8], Avi Ashkenazi[3], Joachim Rudolph [2✉] & Weiru Wang [1✉]

Inositol-Requiring Enzyme 1 (IRE1) is an essential component of the Unfolded Protein Response. IRE1 spans the endoplasmic reticulum membrane, comprising a sensory lumenal domain, and tandem kinase and endoribonuclease (RNase) cytoplasmic domains. Excess unfolded proteins in the ER lumen induce dimerization and oligomerization of IRE1, triggering kinase trans-autophosphorylation and RNase activation. Known ATP-competitive small-molecule IRE1 kinase inhibitors either allosterically disrupt or stabilize the active dimeric unit, accordingly inhibiting or stimulating RNase activity. Previous allosteric RNase activators display poor selectivity and/or weak cellular activity. In this study, we describe a class of ATP-competitive RNase activators possessing high selectivity and strong cellular activity. This class of activators binds IRE1 in the kinase front pocket, leading to a distinct conformation of the activation loop. Our findings reveal exquisitely precise interdomain regulation within IRE1, advancing the mechanistic understanding of this important enzyme and its investigation as a potential small-molecule therapeutic target.

[1] Structural Biology, Genentech, Inc., 1 DNA Way, South San Francisco, CA 94080, USA. [2] Discovery Chemistry, Genentech, Inc., 1 DNA Way, South San Francisco, CA 94080, USA. [3] Cancer Immunology, Genentech, Inc., 1 DNA Way, South San Francisco, CA 94080, USA. [4] Pharmaceutical Development, Genentech, Inc., 1 DNA Way, South San Francisco, CA 94080, USA. [5] Biochemical and Cellular Pharmacology, Genentech, Inc., 1 DNA Way, South San Francisco, CA 94080, USA. [6] BioMolecular Resources, Genentech, Inc., 1 DNA Way, South San Francisco, CA 94080, USA. [7] Microchemistry, Proteomics & Lipidomics, Genentech, Inc., 1 DNA Way, South San Francisco, CA 94080, USA. [8] Proteros Biostructures GmbH, Bunsenstr. 7a, D - 82152 Martinsried, Germany. ✉email: rudolph.joachim@gene.com; wang.weiru@gene.com

The Unfolded Protein Response (UPR) is a highly regulated intracellular network that alleviates endoplasmic reticulum (ER) stress caused by elevated levels of misfolded proteins in the ER lumen[1]. In metazoans, the UPR comprises interplaying signaling branches transmitted through three ER transmembrane proteins: Inositol-Requiring Enzyme 1 (IRE1), Protein kinase R-like Endoplasmic Reticulum Kinase (PERK), and Activating Transcription Factor 6 (ATF6). Of these, IRE1 is the most evolutionarily conserved[2].

IRE1 integrates multiple functional domains into a single ER-membrane-spanning protein. The lumenal domain senses unfolded proteins in the ER through indirect and direct mechanisms: as a result, it undergoes dimerization and subsequent oligomerization[3,4]. The association of the lumenal domain promotes dimerization and oligomerization of the cytoplasmic kinase-RNase tandem domain. The cytoplasmic domain has been shown to self-associate via different surface contacts, where a back-to-back (B2B) configuration is believed to fulfill the essential structural requirements for RNase activation[5,6]. In the B2B dimer, the kinase domains dimerize with their ATP-binding pockets facing away from each other and provide the building blocks for higher-order IRE1 oligomers[5,7]. Furthermore, IRE1 forms distinct supermolecular clusters under certain conditions. These clusters, visualized by fluorescence tagging as foci within cells, are proposed to be the physiologically relevant forms of active IRE1[7].

Evidence so far indicates that the kinase activity of IRE1 is highly directed toward trans-autophosphorylation. The B2B dimeric and oligomeric IRE1 complexes are stabilized by phosphorylation of the kinase activation loop, which in human IRE1 occurs on residues S724, S726, S729, making phosphorylated IRE1 (IRE1-3P) the most active form of the RNase enzyme[8–10].

The IRE1 RNase specifically cleaves the double hairpin motif of unspliced XBP1 mRNA, leading, after ligation of the exons by the RtcB tRNA ligase[11], to a spliced XBP1 (XBP1s) mRNA. The transcript is then translated into a potent transcription factor, which activates numerous genes that help relieve ER stress[12]. Additionally, the IRE1 RNase drives the degradation of a subset of ER-localized mRNAs and microRNAs through an as yet incompletely understood process called Regulated IRE1-Dependent Decay (RIDD)[13]. RIDD is thought to reduce the protein-folding burden on the ER[13], as well as to modulate specific cellular functions, including apoptosis[14,15] and lysosomal function[16].

IRE1 has recently gained interest as a potential therapeutic target[17,18]. Targeting either the kinase or RNase domain with small molecules has been shown to inhibit IRE1 RNase activity[9,19,20]. Uniquely, IRE1 kinase inhibitors are capable of allosterically modulating IRE1's RNase activity in opposite directions. Despite binding with similar affinity to IRE1, most kinase inhibitors allosterically activate the RNase, while some are neutral, and a minority allosterically inhibits the RNase[19,20]. Allosteric activators were shown to stabilize the IRE1 B2B dimer and possibly the oligomer in solution, leading to higher RNase activity[9,10,19,21].

Small-molecule IRE1 modulators provide important tools to probe the physiology of IRE1 in both healthy state and disease. Selective IRE1 RNase activation, in particular, has not been extensively explored, despite recent studies showing that this approach may have therapeutic potential for certain metabolic disorders[22–24], cancers[25], viral infections[26,27], protein aggregation pathologies[28], and neurodegenerative diseases[29]. To date, the best-characterized allosteric activators are APY29[10], IPA[21], and CRUK-3[30]. Of these, only IPA was developed to achieve cellular activity. However, IPA has relatively poor kinase selectivity and activates PERK at low concentrations, possibly inducing

conflicting cellular effects[21]. Better pharmacological tool compounds are needed to further elucidate the principles governing IRE1 RNase activation and its biological implications and therapeutic potential.

During our efforts to identify better allosteric activators, we have discovered G-1749, a potent and very kinase-selective compound that promotes IRE1 RNase activity through a mechanism that involves conformational modification of the kinase activation loop and does not seem to stabilize the B2B dimer. As a consequence, G-1749 stimulates the RNase activity of unphosphorylated IRE1 (IRE1-0P), yet inhibits IRE1-3P. Consistently, it activates pre-associated IRE1 but not the non-associated protein in cells. Our findings reveal a previously unappreciated role of the IRE1 kinase activation loop in regulating RNase activation and open a conceptually distinct path to developing small-molecule activators of IRE1 for potential therapeutic translation.

## Results

**Identification and characterization of allosteric activator G-1749.** We set out to identify allosteric activators of IRE1 RNase by screening internal compound libraries for binding to the kinase, using a published ATP-competitive TR-FRET assay[31]. We then tested kinase binders to assess RNase modulation in an IRE1-specific kinetic RNA cleavage assay that uses the cytoplasmic construct IRE1-LKR (Q470-L977) (Supplementary Fig. 1a). The two most potent activators identified by us, G-9807 and G-1749 (Fig. 1a), bound to the kinase with low-nanomolar $IC_{50}$ values (Supplementary Fig. 1b) and caused RNase activation with similar $EC_{50}$ values (Fig. 1b). Next, we tested these compounds in KMS-11 multiple myeloma cells, known to express functional IRE1 and strongly depend on the IRE1 pathway due to hyper-production of immunoglobulins in the ER[31]. We measured mRNA levels of XBP1s and the well-established RIDD target DGAT2[22] (Fig. 1c and Supplementary Fig. 1c, respectively) using a branched DNA (bDNA) assay. After a 4-h incubation, G-9807 and G-1749 significantly increased the cellular levels of XBP1s mRNA (Fig. 1c) and decreased those of DGAT2 (Supplementary Fig. 1c), indicating compound-induced stimulation of IRE1 RNase activity. We then assessed kinase selectivity and found that G-1749 is a very kinase-selective compound (Supplementary Data 1), inhibiting at 1 μM only 6 of 218 other kinases besides IRE1 by more than 50%. Importantly, neither G-9807 nor G-1749 inhibit PERK (Supplementary Table 1).

G-1749 is a structural analog of AMG-18 (Fig. 1a), a compound that was previously reported[32] and confirmed[31] to be a potent and selective allosteric inhibitor of the IRE1 RNase.

AMG-18 and G-1749 showed similar kinase binding $IC_{50}$ values (Supplementary Fig. 1b), but exerted opposite modulation of RNase activity, both in the RNA cleavage assay (Fig. 1b) and in KMS-11 cells (Fig. 1c, Supplementary Fig. 1c). G-1749 and AMG-18 are structurally very similar, except for different substituents on the naphthyl moiety henceforth referred to as tail. Intrigued by this close structural similarity and the nonetheless opposite RNase modulation, we set out to characterize G-1749 further.

**G-1749 does not stabilize IRE1 dimers in solution.** Previously identified IRE1 RNase allosteric activators were shown to do so by stabilizing IRE1 dimers and even oligomers in solution[10,21]. This effect resembles the stabilization effect brought about by phosphorylation of the IRE1 kinase activation loop on S724, S726, and S729[8,9].

To examine whether the activators identified by us affect the oligomeric state of IRE1, we used sedimentation velocity analytical ultracentrifugation (SV-AUC) (Fig. 1d, e). IRE1-0P

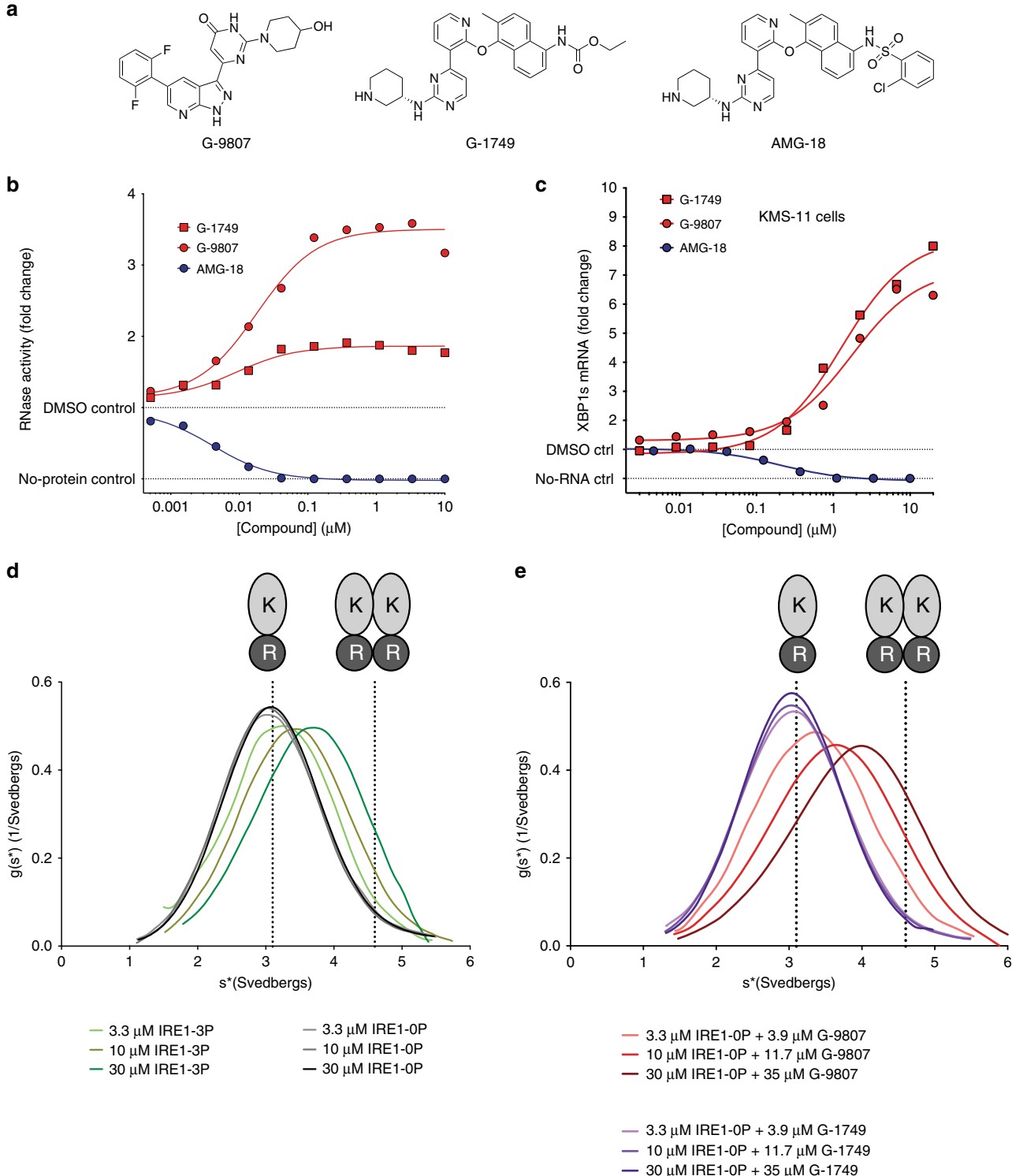

**Fig. 1 Characterization of IRE1 kinase inhibitors. a** Chemical structures of IRE1 kinase (K) inhibitors. **b** IRE1 RNase (R) activity in the presence of compounds in panel **a**. RNA cleavage was measured by a kinetic fluorescence assay using an XBP1-like RNA hairpin substrate and IRE1 LKR construct (Q470-L977). Background from the no-protein control was subtracted from signal before calculating fold change. Source data are provided as a Source Data file. Data are presented as the mean for measurements from two independent experiments ($n = 2$). **c** RNA levels for IRE1 RNase product XBP1s measured by bDNA assay of lysates from KMS-11 cells treated with compounds for 4 h. Background from the no-RNA control was subtracted from signal before calculating fold change. Source data are provided as a Source Data file. Data are presented as the mean for measurements from two independent experiments ($n = 2$). **d, e** Analytical Ultracentrifugation sedimentation velocity (SV-AUC) experiments for (**d**), IRE1 LKR (Q470-L977) unphosphorylated and autophosphorylated and (**e**), IRE1 LKR unphosphorylated in the presence of slight excess G-9807 and G-1749. Calculated sedimentation coefficients for IRE1 monomer (3.1 svedbergs) and dimer (4.6 svedbergs) are highlighted by dotted lines. Source data are provided as a Source Data file.

**Table 1 X-ray crystallography data collection and refinement statistics.**

|  | IRE1-0P/G-9807 | IRE1-0P/G-1749 | IRE1-0P/G-7658 | IRE1-3P/cpd.2 | IRE1-0P apo | IRE1-3P apo |
|---|---|---|---|---|---|---|
| **Data collection** |  |  |  |  |  |  |
| Space group | C2 | $P2_12_12_1$ | $P2_12_12_1$ | $P2_1$ | $C222_1$ | $P2_1$ |
| Cell dimensions |  |  |  |  |  |  |
| $a, b, c$ (Å) | 73.51, 112.76, 67.89 | 77.71, 82.92, 164.62 | 77.11, 82.39, 164.03 | 61.67, 139.24, 63.95 | 68.04, 167.59 103.49 | 47.49, 158.66, 155.94 |
| $\alpha, \beta, \gamma$ (°) | 90.00, 119.58, 90.00 | 90.00, 90.00, 90.00 | 90.00, 90.00, 90.00 | 90.00, 114.90, 90.00 | 90.00, 90.00, 90.00 | 90.00, 91.01, 90.00 |
| Wavelength (Å) | 1.00000 | 1.00000 | 0.97946 | 0.97946 | 0.97949 | 0.97949 |
| Resolution (Å) | 55.61–2.08 (2.12–2.08)[a] | 82.31–1.74 (anisotropic) 82.31–2.05 (isotropic) (1.94–1.74) | 58.12–2.61 (anisotropic) 58.12–3.05 (isotropic) (2.84–2.61) | 55.94–2.74 (anisotropic) 55.94–3.02 (isotropic) (2.97–2.74) | 44.03–2.57 (2.58–2.57)[a] | 47.48–2.3 (2.31–2.30) |
| $R_{merge}$ | 0.062 (0.762) | 0.065 (1.067) | 0.148 (1.227) | 0.173 (0.798) | 0.064 (1.155) | 0.066 (0.665) |
| $I / \sigma(I)$ | 12.2 (1.12) | 16.6 (1.7) | 10.4 (1.5) | 5.6 (1.2) | 23.2 (2.0) | 12.5 (2.0) |
| CC(1/2) | 0.99 (0.59) | 0.99 (0.70) | 0.99 (0.73) | 0.975 (0.355) | 0.99 (0.74) | 0.99 (0.71) |
| Completeness | 97.8 (90.3) | 95.2 (ellipsoidal) 70.5 (spherical) | 91.5 (ellipsoidal) 77.5 (spherical) | 85.4 (ellipsoidal) 67.4 (spherical) | 99.8 (98.0) | 99.5 (97.9) |
| Redundancy | 2.9 (2.7) | 6.6 (6.4) | 6.7 (7.0) | 3.0 (2.8) | 9.8 (10.0) | 3.8 (3.7) |
| **Refinement** |  |  |  |  |  |  |
| Resolution (Å) | 55.61–2.08 | 82.31–1.74 (anisotropic) | 58.12–2.61 (anisotropic) | 55.94–2.74 (anisotropic) | 44.03–2.57 | 47.48–2.3 |
| No. reflections | 26523 | 76917 | 25309 | 17403 | 36280 | 195063 |
| $R_{work}/R_{free}$ | 0.191/0.217 | 0.195/0.233 | 0.247/0.290 | 0.262/0.317 | 0.203/0.246 | 0.206/0.248 |
| No. atoms |  |  |  |  |  |  |
| Protein | 3224 | 11698 | 10835 | 11682 | 3204 | 13005 |
| Ligand/ion | 31 | 76 | 76 | 100 |  | 120 |
| Water | 128 | 389 | 42 |  | 17 | 353 |
| B-factors |  |  |  |  |  |  |
| Protein | 61.89 | 48.47 | 70.18 | 85.32 | 70.81 | 53.32 |
| Ligand/ion | 48.66 | 32.00 | 29.00 | 81.00 |  | 68.00 |
| Water | 54.24 | 45.39 | 45.18 |  | 60.58 | 53.43 |
| R.m.s. deviations |  |  |  |  |  |  |
| Bond lengths (Å) | 0.005 | 0.008 | 0.003 | 0.003 | 0.009 | 0.003 |
| Bond angles (°) | 0.731 | 1.101 | 0.646 | 0.777 | 1.148 | 0.677 |

[a]Values in parentheses are for highest-resolution shell.

was predominantly monomeric up to a concentration of 30 μM, while phosphorylation partially stabilized the IRE1 dimer as expected (Fig. 1d), with a calculated dimerization $K_d = 67$ μM (Supplementary Fig. 2a). The relatively weak self-association affinity indicates a tendency of the recombinant cytoplasmic IRE1 domains to rapidly associate and dissociate in solution in absence of the lumenal and transmembrane domains. Upon incubation of IRE1-0P with a slight excess (ca. 15% molar excess) of G-9807, the equilibrium shifted towards dimeric IRE1, similarly to what was observed previously for allosteric RNase activators, with a calculated $K_d = 35$ μM (Fig. 1e, Supplementary Fig. 2b). In contrast, incubation of IRE1-0P with slight excess of allosteric inhibitor AMG-18 did not stabilize dimers (Supplementary Fig. 1d). Surprisingly, RNase activator G-1749 also did not increase the dimeric population (Fig. 1e). These results suggest that while G-9807 acts as an allosteric RNase activator by stabilizing dimerization, G-1749 operates through a distinct mode, without altering the monomer-dimer equilibrium of IRE1. To seek further mechanistic insight into the different RNase regulation modes of these compounds, we solved the co-crystal structure of each compound with IRE1.

**G-1749 occupies a pocket proximal to the DFG motif.** We determined the crystal structures of G-9807 and G-1749 in complex with IRE1-0P at about 2 Å resolution (Table 1). The

structures are shown in Fig. 2 together with the structures of AMG-18-bound-[31] and apo IRE1, including the first reported crystal structure of human apo IRE1-3P. See Supplementary Fig. 10 for Fo-Fc omit maps of all compounds.

All structures show IRE1 as a B2B dimer (Fig. 2a), with extensive overlay of the RNase domains, but some differences in the kinase domains, especially in the conformation of the activation loop and αEF-αF loop. The compounds occupy the kinase ATP-binding pocket, forming key hydrogen bonds with the kinase hinge motif, specifically the backbone of C645. G-9807 also forms hydrogen bonds with E643 at the hinge, and H692 in the front of the pocket, while G-1749 and AMG-18 interact with the sidechain of E651. Additionally, all compounds contact key structural elements in the ATP-binding pocket, including the conserved Lys-Glu salt bridge (K599-E612) and the DFG motif (D711, F712, G713): G-9807 and AMG-18 bind directly to the catalytic lysine (K599) in the kinase salt bridge, while G-1749 binds to the salt bridge counterpart E612. AMG-18 interacts with F712 in the DFG motif, while both G-1749 and AMG-18 bind to the backbone of D711 in the motif (Fig. 2c).

The two RNase allosteric activators induce protein conformations typical of an active kinase in complex with a type 1 kinase inhibitor and also evident in apo structures: both the DFG motif and the C-helix are in the "in" conformations, and the salt bridge between K599 and E612 is intact (Fig. 2c). Conversely, the chlorobenzenesulfonamide tail of allosteric inhibitor AMG-18 disrupts

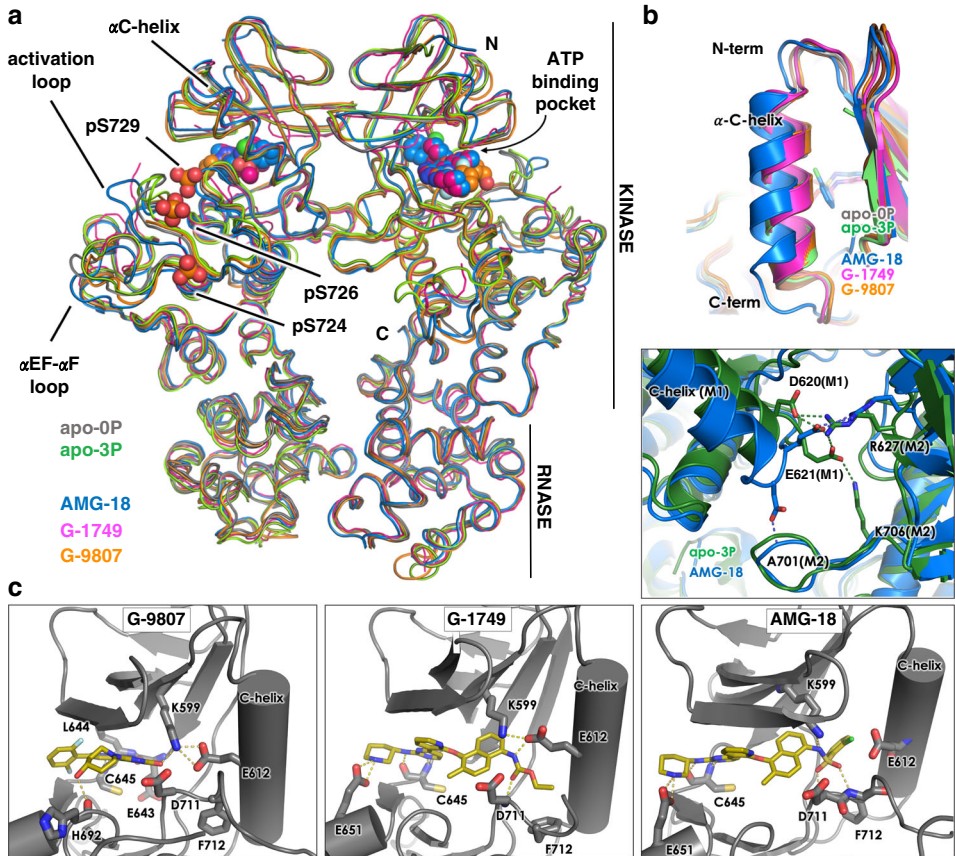

**Fig. 2 Crystal structures of IRE1 RNase activators G-9807 and G-1749, and inhibitor AMG-18 in complex with unphosphorylated IRE1. a** Overall alignment of B2B IRE1 dimers for each crystal structure shown as ribbons in colors as indicated by legend. Ligands are shown as spheres. Phosphorylated serine residues in the IRE1-3P structure are shown as sticks and spheres. **b** Top, C-helix conformation for all determined structures. Bottom, details of B2B dimer interface mediated by the C-terminus of the C-helix for IRE1 apo 3P and AMG-18/IRE1-0P structures. Protein shown as cartoon, with key residues and ligands shown as sticks. Colors as in legend. **c** Details of compound interactions in the ATP-binding pocket. Protein is shown as cartoon with key residues shown as sticks. Ligands shown as sticks.

the K599-E612 salt bridge, allowing the C-helix to swing outward and the tail benzene ring to occupy the back pocket (Fig. 2b, c). It has been shown that most type 1 kinase inhibitors of IRE1 activate the RNase activity of IRE1[20], while an outward swing of the C-helix is likely to disrupt the productive B2B dimer, leading to allosteric RNase inhibition[32]. Inconsistent with findings from our SV-AUC studies, IRE1 co-crystallized as B2B dimer in the presence of AMG-18 (Fig. 2a). This likely results from the high protein concentration (ca. 200 μM) in the crystallization conditions. Nevertheless, the C-helix in the AMG-18-bound structure not only is in a more outward position, but it also shows partial unwinding of its C-terminal end, where key residues of the B2B dimer interface D620 and E621 are located, leading to a modified hydrogen bond pattern in the dimer interface (Fig. 2b). In IRE1 apo structures and IRE1 in complex with allosteric activators, D620 and E621 of one monomer form salt bridges with R627 and K706, respectively, of the second monomer, while in the AMG-18 structure D620 has flipped outward due to the helix unwinding, hindering hydrogen bond formation with K706 and interacting instead with the backbone of A701. This alternative hydrogen bond network is likely to weaken the dimer interface, consistent with the SV-AUC experiment described above and the observed inhibitory effect of AMG-18.

One key feature observed in the G-1749 structure is the conformation of the kinase DFG motif and activation loop: rather than occupying the back pocket as the benzene sulfonamide of AMG-18, the ethyl carbamate tail of G-1749 wedges itself

underneath the C-helix, in proximity to the DFG motif, creating a front pocket (Figs. 2c and 3a). This pocket is formed mainly by a 180-degree flip of F712, and is flanked by E612, L615, and L616 on the C-helix, D711 and G713 of the DFG motif, and L714 in the activation loop. A similar conformation of the DFG motif and the loop has been observed before in IRE1 face-to-face (F2F) dimer crystal structures (Supplementary Fig. 3)[33], but no F2F interaction is present in the crystal lattice of the G-1749 complex. Additionally, in F2F dimers, the N-terminus of the C-helix associates with part of the activation loop into a β-strand, a feature that is absent in the complex with G-1749. By occupying this front pocket, the tail spares the K599-E612 salt bridge and does not insult the C-helix, perhaps explaining why G-1749 activates the RNase instead of inhibiting it.

To investigate whether this peculiar conformation had any role in the mechanism of activation of G-1749, we evaluated different tails to substitute the ethyl carbamate on the naphthyl moiety (Fig. 3, Supplementary Table 2). In the biochemical RNase assay (Fig. 3b, c, left panels) we found that less bulky substituents like buten- or butynamide (compound **1** and G-7658, respectively) led to higher RNase activation, while larger substituents such as branched and longer carbamates or amides (compounds **6-12**) led to inhibition. This is likely due to clashes with residues lining the front pocket, which in turn would force the tails to occupy the back pocket and break the salt bridge, thereby causing RNase inhibition. Interestingly, a methyl carbamate tail (compound **2**), or the truncation of the tail down to aniline (compound **3**), led to

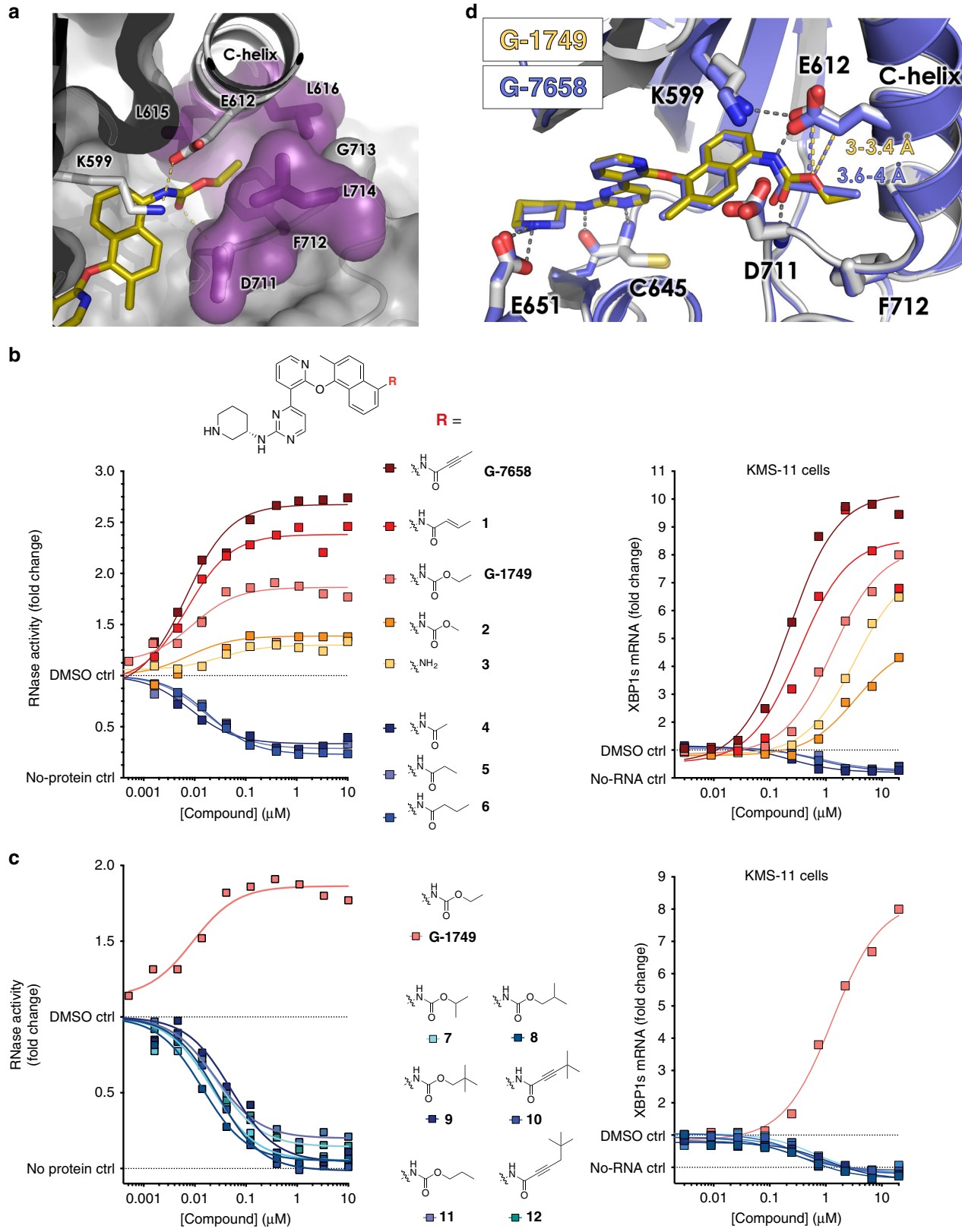

lower activation than G-1749 in the biochemical assay (Fig. 3b, left panel), suggesting that a complete stabilization of the front pocket is important to achieving robust activation. Experiments in KMS-11 cells (Fig. 3b, c, right panels) confirmed the effect of these tail modifications on IRE1 RNase activation. One outlier

was compound **3**, which showed very low activation in biochemical settings, similarly to compound **2**, but induced higher levels of XBP1s RNA in KMS-11 cells than compound **2**. The chemical substituents tolerated by the front pocket seem to be very limited: even a minor substitution of the carbamate

**Fig. 3 The nature of the tail substituent determines the degree of RNase activation. a** Co-crystal structure of G-1749 (golden sticks) in complex with unphosphorylated IRE1 (gray cartoon). Details of the front pocket where G-1749 ethyl carbamate tail binds are highlighted as magenta sticks and surface. **b, c** Left, dose-response curves in biochemical RNA cleavage assay; right, XBP1s mRNA levels measured by bDNA assay after treatment with compounds for 4 h, for indicated derivatives of G-1749. In all experiments background from the no-protein and no-RNA controls were subtracted from signal before calculating fold change. Source data are provided as a Source Data file. Data are presented as the mean for measurements from two independent experiments ($n = 2$). **d** Co-crystal structure of G-7658 (purple sticks) in complex with unphosphorylated IRE1 (purple cartoon) overlayed on the co-crystal structure of G-1749 (golden sticks) in complex with unphosphorylated IRE1 (gray cartoon). Conserved hydrogen bonds are shown as gray dashes. The distances between E612 and the oxygen of G-1749 carbamate tail or the alkynyl carbon of G-7658 tail are highlighted as golden or purple dashes, respectively, and the measured distance is reported in the corresponding color.

oxygen atom to a larger methylene, from ethyl carbamate to propyl amide (compound **6**), or from methyl carbamate to methyl or ethyl amide (compound **4** or **5**, respectively), or increasing the tail length by one atom, from ethyl to propyl carbamate (compound **11**), is enough to impart a switch from activation to inhibition (Fig. 3b, c). We then co-crystallized a more potent allosteric RNase activator from our SAR effort, G-7658, with IRE1-0P and found that it superimposes perfectly with G-1749 (Table 1, Fig. 3d), except for an increase of approximately 0.5 Å in distance from E612.

**G-1749 modifies the conformation of the IRE1 kinase activation loop.** The co-crystal structure of G-1749 with IRE1 lacks electron density data for the kinase activation loop from C715 to P732 and from E745 to K748 in the αEF-αF loop. The available density strongly suggests that the activation loop is adopting a different conformation from the one found in most IRE1 structures. In fact, it seems that the creation of the front pocket by G-1749 selects for an upward loop conformation, as observed in F2F IRE1 dimer structures (Supplementary Fig. 3).

Using our crystal structure as template and the sequence for the missing loops, we used the Molecular Operating Environment (MOE)[34] software to generate models of the loops (Fig. 4a and Supplementary Fig. 4). The models show upward conformations of the activation loop, different from the one previously observed for active IRE1 (Fig. 4a). The section of the activation loop between L714 and P732 seems to favor bending towards the G-loop, placing itself up to 40 Å away from the normally extended loop conformation. The αEF-αF loop seems to have filled the space left unoccupied by the activation loop, moving upwards about 8 Å. The absence of density for these regions suggests that they may be highly dynamic in the G-1749-IRE1 complex, but, nonetheless, upward conformations seem to be favored by the opening of the front pocket in the presence of G-1749.

To study IRE1 conformation in solution, we performed hydrogen-deuterium exchange (HX-MS) experiments (Fig. 4b-d) comparing apo IRE1-0P or IRE1-3P to IRE1-0P in complex with G-1749, G-9807, or AMG-18. A key finding from these studies was that peptide R687DLKPHNIL showed much faster deuteration in the presence of G-1749, compared to G-9807 or AMG-18 (Fig. 4b). This peptide belongs to the kinase catalytic loop containing the HRD motif, which, in B2B IRE1 dimer structures, is closely packed against the activation loop (Fig. 4d, compare G-9807 to G-1749). We calculated protection factors (pF) for the region[35] for each condition compared to IRE1-0P apo (Fig. 4c). The sample containing G-1749 showed $0 < pF < 1$, corresponding to destabilization. Conversely, G-9807 conferred protection from deuteration to the region (pF > 1), similarly to IRE1-3P. Destabilization of the area in the presence of G-1749 could be explained by an upward loop conformation leading to weakened hydrogen bonding. By contrast, the presence of phosphorylations or G-9807 seems to stabilize an extended loop conformation that protects the region from deuteration, as shown in the corresponding crystal structures.

**G-1749 allosterically inhibits the RNase activity of phosphorylated IRE1.** Our results suggest that the activation loop conformation is critical to the mechanism of allosteric RNase modulation by G-1749. As autophosphorylation of IRE1 on the activation loop is important for kinase-mediated regulation of IRE1 RNase activity, we wondered how phosphorylation would impact the mechanism of G-1749.

We used a shorter protein construct, IRE1-KR (G547-L977, see Supplementary Fig. 1a), to reduce the complexity associated with autophosphorylation of the N-terminal linker region by the IRE1 kinase (see "Methods" and Supplementary Table 3).

Consistent with our previous data with IRE1-LKR (Fig. 1b), G-1749, and G-9807 significantly increased RNase activity of IRE1-KR-0P, while AMG-18 inhibited it (Fig. 5a). G-9807 also increased activity of IRE1-KR-3P (Fig. 5a), but to a much lesser degree than for IRE1-0P. G-1749 in this setting actually inhibited RNase activity, as did AMG-18 (Fig. 5a). SV-AUC experiments with AMG-18 and G-1749 in complex with IRE1-3P showed that both compounds inhibit dimer formation (Fig. 5b), suggesting that the tail of G-1749 likely occupies the back pocket of IRE1-3P, thereby disrupting the C-helix position and leading to RNase inhibition. Of note, all compounds derived from G-1749 (Fig. 3) also inhibited IRE1-3P (Supplementary Fig. 5), albeit to different degrees. More potent allosteric RNase activators of IRE1-0P, such as G-7658, inhibited IRE1-3P only partially, suggesting that perhaps certain compounds experience mixed binding modes in the presence of phosphorylation, where their tail can be accommodated in either the front or back pocket. To assess this, we solved the co-crystal structure of compound **2** in complex with IRE1-3P (Table 1, Fig. 5c). The methyl carbamate tail of **2** is occupying the back pocket of IRE1-3P and breaking the K599-E612 salt bridge, in a binding mode that is entirely similar to that of allosteric RNase inhibitor AMG-18 in complex with IRE1-0P.

The inhibition of IRE1-3P by G-1749 suggests that activation loop phosphorylation is incompatible with stable opening of the front pocket. A possible reason for this is the stabilization of the extended activation loop conformation by the extensive hydrogen bonding network that the phosphates on S724, S726, and S729 form with the backbone and sidechains on the activation loop (Fig. 5d). The resulting rigidification would make the opening of the front pocket, which would require an upward shift of the activation loop, energetically unfavorable, depending perhaps on the number of phosphates present on the loop. To examine this further, we substituted S724, S726 and S729 to alanine residues, in singlets or pairs. We then generated the fully autophosphorylated forms of these mutants (2P for single mutants, 1P for double mutants, see Supplementary Table 3 for phosphorylation site mapping). RNA cleavage experiments revealed that phosphorylation of one serine on the activation loop does not appreciably increase RNase activity (Supplementary Fig. 6a), while two phosphorylation events, and especially the combination pS724 + pS726, do augment RNA cleavage in our assay (Supplementary Fig. 6b). Nevertheless, IRE1-3P was more active than any IRE1-2P variant. G-9807 was capable of activating all 1P and 2P mutant

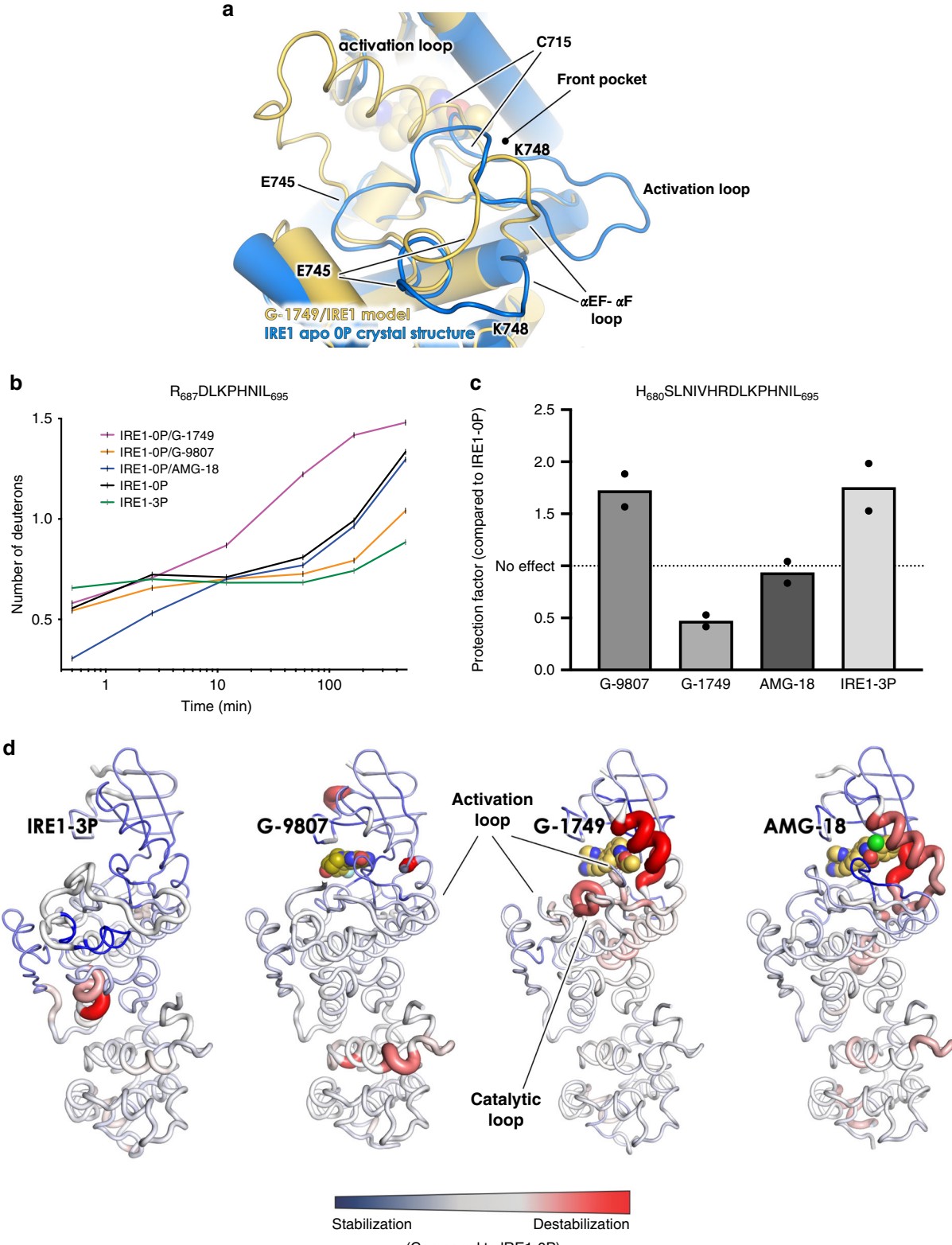

variants (Supplementary Fig. 7a), whereas AMG-18 fully inhibited the active mutants (Supplementary Fig. 7b). G-1749 dose-dependently activated 1P mutants, and partially inhibited 2P variants (Fig. 5e). To pinpoint the exact interactions responsible for this phenomenon, we mutated the phosphate-interacting residues on the loop to alanine, as single or multiple combinations, and produced the corresponding fully phosphorylated proteins (3P) (See Methods, and Supplementary Table 3 for phosphorylation site mapping). G-1749 partially inhibited the majority of these mutants, but was neutral to the R687A-3P mutant, and, more importantly, was capable of activating R687A/R611A-3P, R687A/K716A-3P, and the triple combination R687A/

**Fig. 4 G-1749 modifies the conformation of the IRE1 kinase activation loop. a** Model of activation loop and αEF-αF loop segments (yellow cartoon) of the IRE1 kinase for which electron density was missing in the G-1749/IRE1 co-crystal structure, built using the crystal structure model (Chain A) as template, using the Homology Model algorithm in the Molecular Operating Environment (MOE) software, overlaid with IRE1 apo 0P crystal structure (blue cartoon). The model shown here was the one reported as optimized among ten determined by the program (see Supplementary Fig. 4). **b** Hydrogen–deuterium exchange (HX-MS) results for peptide R687-L695 in indicated samples, shown as number of deuterons over time. Samples were incubated in deuterated buffer from 30 s up to 8 h and then subjected to protease digestion after quenching. Peptides were separated by HPLC, and mass spectrometry was utilized to measure deuteration levels of peptides at the indicated timepoints. The peptide from the sample with IRE1-0P in the presence of G-1749 showed a much faster deuteration than IRE1-0P apo and in complex with AMG-18. Source data are provided as a Source Data file. Data are presented as the mean for measurements from two independent experiments ($n = 2$). **c** Data from panel **b** were used to calculate protection factors, pF (see Methods and SI), for indicated samples, compared to IRE1-0P. pF above a value of 1 indicate protection/stabilization, while pF between 0 and 1 indicate deprotection/destabilization of the region. Source data are provided as a Source Data file. Data are presented as the mean for measurements from two independent experiments ($n = 2$). **d** pF for all detected peptides are reported on the corresponding structures as cartoon putty, with thicker cartoon corresponding to larger values, and vice versa. Stabilization (pF > 1) is shown as blue, destabilization (0 < pF < 1) is shown as red. No change is shown as gray. Source data are provided as a Source Data file.

R611A/K716A-3P (Fig. 5f). This suggests that interaction of HRD motif's R687 with pS729 is a key limitation to the ability of G-1749 to open and stabilize the front pocket for RNase activation. Removing this interaction, alone or in combination with the other hydrogen bonds to pS729, allows G-1749 to modify the activation loop conformation as it does with IRE1-0P.

**G-1749 activates pre-associated IRE1 in a cellular context.** Although G-1749 seems to effect conformational changes of the kinase activation loop of IRE1-0P, it is unclear precisely how this leads to RNase activation. Our initial hypothesis was that G-1749 allosterically modifies a potential direct communication corridor between the kinase and RNase domains. A defined chain of conserved residues connects the two enzymes (Supplementary Fig. 8), including P830, right at the domain interface. Supporting the possibility of allosteric interdomain communication is that a naturally occurring P830L mutation inactivates the RNase and severely impairs kinase activity[31,32]. However, the conserved residues in the connecting region do not show any significant shift upon alignment of the G-1749 structure with our other IRE1 structures, either apo, or co-crystallized with other ligands, with the exception of the DFG motif, as discussed (Figs. 2a, c, 3a). We therefore turned to our HX-MS experiments to interrogate changes in structural dynamics. Generally, G-9807 protected the IRE1 kinase from deuteration for all observed peptides (Fig. 4d and Supplementary Data 2). A similar stabilization effect was observed in the IRE1-3P sample. This suggests that stabilization of the activation loop in an extended conformation by phosphorylation or binding to allosteric RNase activators may stabilize IRE1 B2B dimers through widespread rigidification of the kinase in the monomer context. Conversely, G-1749 did not display the same ability, showing instead a certain degree of destabilization of the kinase, perhaps due to the highly dynamic nature of the activation loop in the presence of G-1749 (Fig. 4d and Supplementary Data 2). RNase domain regions did not display any differences in deuteration pattern among any samples containing different compounds, with the exception of some destabilization observed at the C-terminus of the domain in the G-9807 and AMG-18 samples. These data suggest that G-1749 does not augment activity through induction of allosteric conformational changes in the RNase domain, at least in the context of an isolated monomer, which is the likely state of IRE1 in the HX-MS experiment (protein concentration is about 4 μM during labeling).

To seek further insight into the activation mechanism of G-1749, we tested this molecule and G-9807 in cancer cell lines that we previously characterized for IRE1 pathway activity, i.e., KMS-11 multiple myeloma cells[31], MDA-MB-231 triple-negative breast cancer cells (Fig. 6).

G-9807 activated the IRE1 pathway in both cell lines, as indicated by Western blot analysis of XBP1s levels (Fig. 6a, b). G-1749 activated the pathway in KMS-11 cells (Fig. 6a), but induced no detectable activation in MDA-MB-231 cells at concentrations of up to a 1 μM (Fig. 6b). A key difference between these two cell lines is that the IRE1 pathway is constitutively activated in KMS-11 cells due to hyper-production and frequent misfolding of immunoglobulins in the ER, but not in MDA-MB-231 cells. The constitutive IRE1 activation in KMS-11 cells is accompanied by constitutive IRE1 association, as confirmed by disuccinimidyl suberate (DSS) crosslinking of cell lysates (compare lane 1 of each blot of Fig. 6a, b). Furthermore, the levels of IRE1 protein are higher in KMS11 than in MDA-MB-231 cells (Fig. 6c); as a corollary, in KMS-11 cells IRE1 dimers/oligomers are already present at baseline, whereas in MDA-MB-231 cells they appear only after exogenous induction of ER stress using Thapsigargin (Fig. 6a, b). Consistent with its ability to induce IRE1 dimerization, G-9807 switched on the IRE1 pathway in MDA-MB-231 cells, as indicated by an increase in IRE1 dimerization and in XBP1s levels compared to the DMSO control. By contrast, G-1749 did not seem capable of inducing IRE1 dimerization and appeared to boost IRE1 activity only in the cell line with pre-associated IRE1, i.e., KMS-11. One strategy to induce spontaneous IRE1 association is to increase the protein's cellular levels: a high IRE1 concentration in the ER membrane leads to stress- and phosphorylation-independent IRE1 association, which in turn translates into low-level production of XBP1s. To verify that G-1749 can activate pre-associated IRE1, we transfected MDA-MB-231 cells harboring a CRISPR/Cas9 knockout of IRE1 (IRE1$^{-/-}$) with an expression plasmid encoding IRE1 cDNA under control of the cytomegalovirus (CMV) promoter. To avoid potential complications due to hyperphosphorylation of the overexpressed protein, we used a kinase-dead mutant of IRE1, D688N, in place of WT. This enabled abundant IRE1 protein expression, which in turn led to spontaneous IRE1 dimerization, with little XBP1 splicing (Fig. 6c, right gel, and Fig. 6d, second lane). Importantly, addition of G-1749 augmented RNase activity of D688N IRE1 as indicated by XBP1s production, with similar potency to that in KMS-11 cells (Fig. 6d). G-9807 also increased RNase activity, while AMG-18 did not.

Together, these results suggest that G-1749 is capable of activating IRE1 only when the protein is pre-associated, unlike G-9807, which can switch on IRE1 by promoting its self-association.

## Discussion

In this work, we have identified a class of highly potent and selective IRE1 kinase inhibitors that allosterically activate the IRE1 RNase, as represented by G-1749. There are two key features that differentiate this class of activators from previously

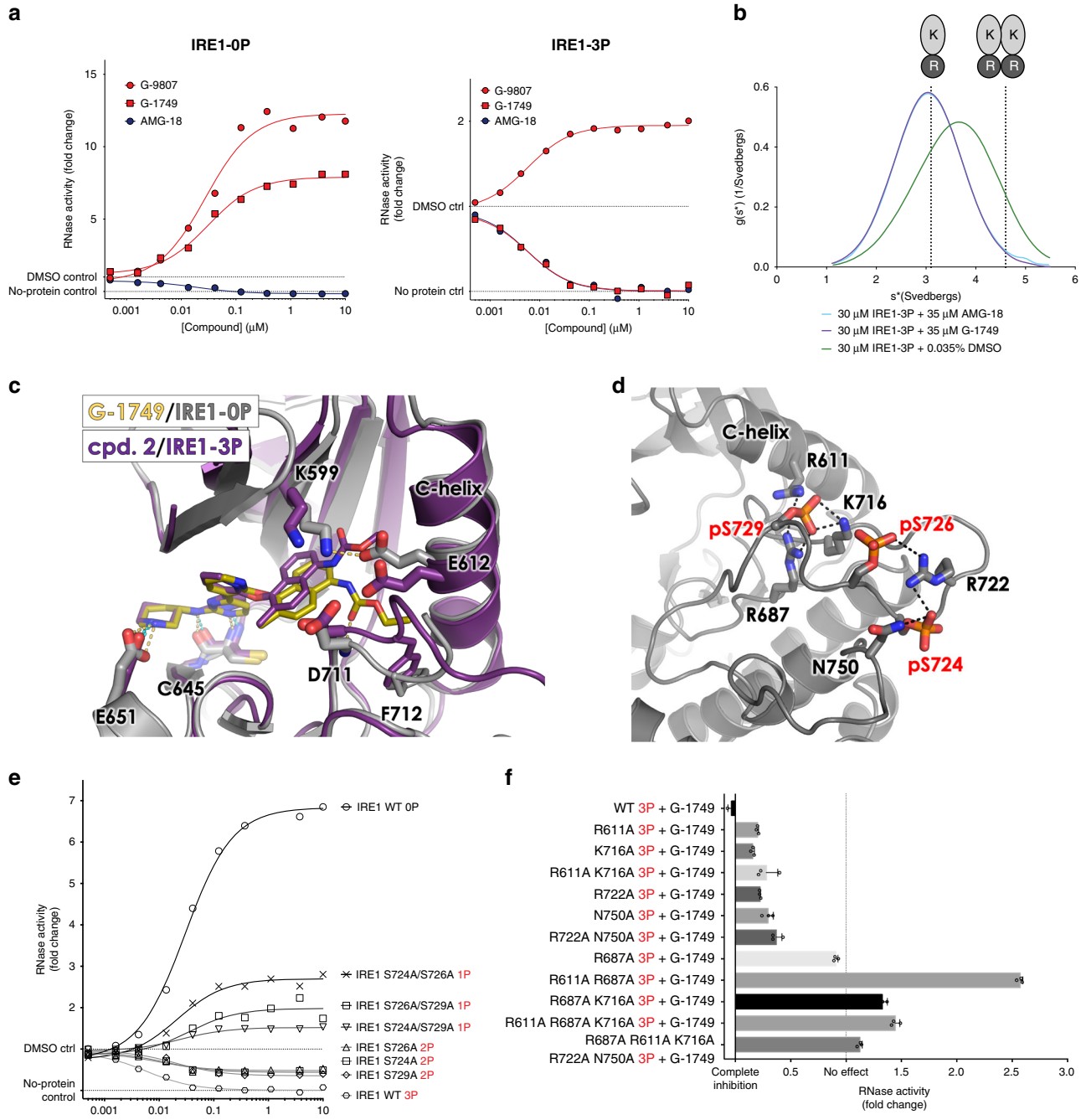

**Fig. 5 G-1749 modulation of IRE1 RNase is dependent on the kinase phosphorylation state. a** IRE1 RNase activity in the presence of AMG-18, G-9807, or G-1749. RNase activity was measured by the same assay as Fig. 1, but using an IRE1 KR construct (G547-L977) unphosphorylated (0P, left), or fully phosphorylated (3P, right) on S724, S726, S729. Source data are provided as a Source Data file. Data are presented as the mean for measurements from two independent experiments ($n = 2$). **b** (SV-AUC) experiments for IRE1 LKR autophosphorylated alone or in the presence of slight excess G-1749 or AMG-18. Source data are provided as a Source Data file. **c** Co-crystal structure of **2** (purple sticks) in complex with IRE1-3P (purple cartoon) aligned with co-crystal structure of G-1749 (gold sticks) in complex with IRE1-0P (gray cartoon). **d** Details of the IRE1 kinase activation loop from the crystal structure of apo IRE1-3P (green cartoon). Phosphorylated serines and interacting residues are highlighted in sticks. **e** RNase activity for IRE1 KR phosphoserine S/A mutants in the presence of increasing concentration of G-1749. 1P or 2P refers to the phosphorylation state of each mutant, as verified by intact mass spectrometry and phosphorylation mapping after protein digestion (see "Methods" and SI). Background from the no-protein control was subtracted from signal before calculating fold change. Source data are provided as a Source Data file. Data are presented as the mean for measurements from two independent experiments ($n = 2$). **f** RNase activity in the presence of 10 μM G-1749 for fully phosphorylated IRE1 KR mutants of residues that would interact with phosphoserines, as in (**d**). Background from the no-protein control was subtracted from signal before calculating fold change. Source data are provided as a Source Data file. Data are presented as the mean and S.D. for measurements from three independent experiments ($n = 3$).

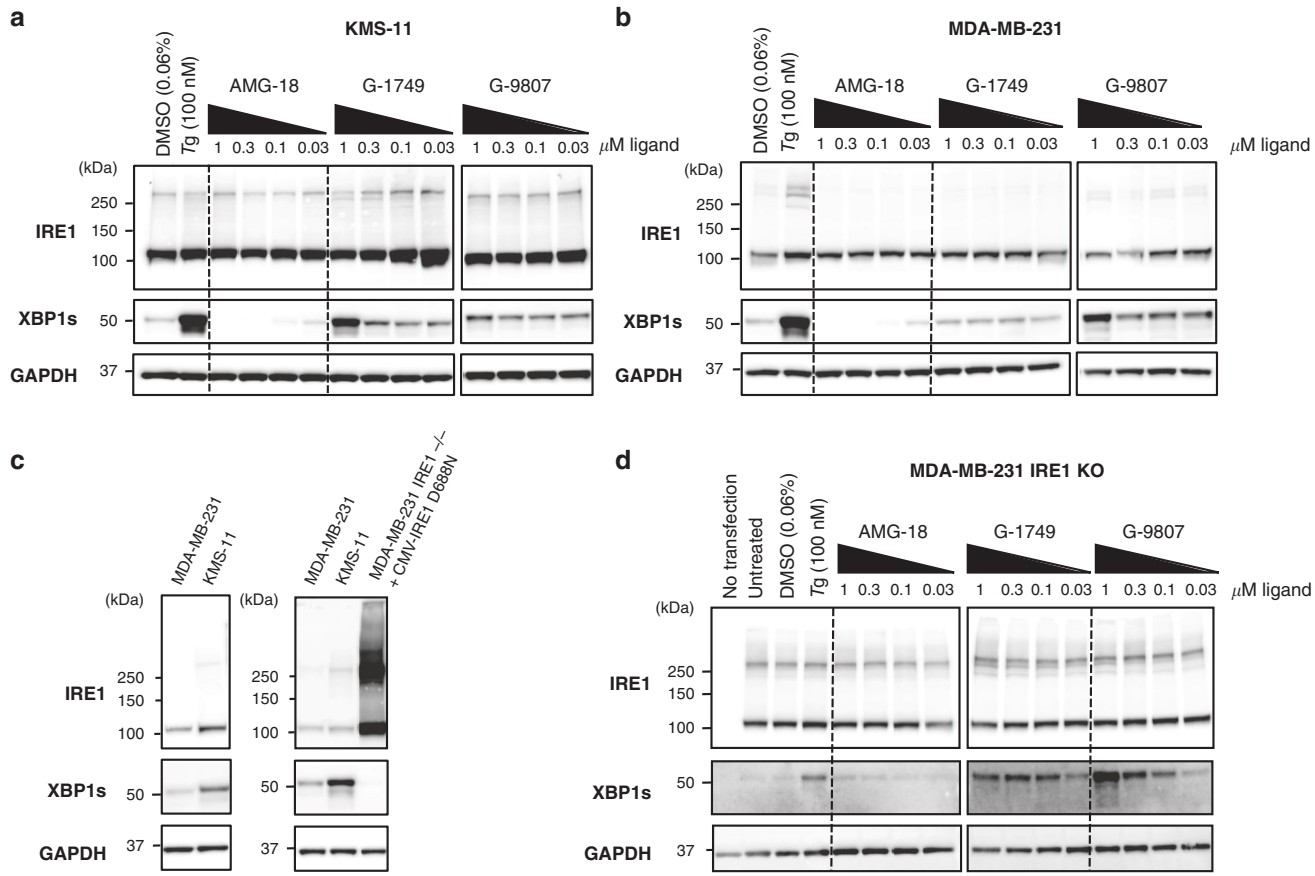

**Fig. 6 G-1749 acts on pre-associated IRE1. a, b** Western blots for indicated proteins of KMS-11 (**a**) and MDA-MB-231 (**b**) cell lysates after treatment of the cells for 4 h with indicated compounds. DMSO was used as a negative control (0.06%) and Thapsigargin (Tg, 100 nM) as a positive control. Before being loaded on a gel, lysates were crosslinked with disuccinimidyl suberate (DSS, 250 nM) for 1 h at room temperature, quenched with TRIS buffer (1 M) for 15 min at room temperature. One representative out of two experiments is shown. **c** Comparison of IRE1 and XBP1s protein concentration in, left, KMS-11 and MDA-MB-231 cell lines, compared to, right, MDA-MB-231 IRE1−/− after transfection with IRE1 kinase-dead (D688N) using a CMV promoter. **d** Kinase-dead (D688N IRE1) under the control of a CMV promoter was overexpressed in MDA-MB-231 IRE1 KO cells using a lipofectamine kit. Transfected cells were then treated with compounds for 4 h. Reported are western blots of indicated proteins after crosslinking with DSS. One representative out of two experiments is shown. Source data for all blots are provided as a Source Data file.

reported IRE1 modulators. First, the exquisite kinase selectivity precludes interference with the other kinase member of the UPR, PERK (Fig. 1, Supplementary Data 1, Supplementary Table 1), therefore ensuring that cellular effects reflect modulation of the IRE1 branch. Second, G-1749 occupies a front pocket in proximity to the DFG motif that was previously only observed in F2F IRE1 dimer structures and never in B2B dimers (Figs. 2c, 3a, Supplementary Fig. 3). By doing so, G-1749 appears to force the activation loop into an upward conformation, which is consistent with HX-MS data obtained by us (Fig. 4 and Supplementary Fig. 4). Our efforts also led to the discovery of more traditional IRE1 activators such as G-9807 (Fig. 1a). G-9807 acts similarly to previously published activators and to IRE1 autophosphorylation through stabilization or induction of dimers (Fig. 1d, e). In contrast, G-1749 does not seem to stabilize or induce IRE1 dimers (Fig. 1e). The evidence for IRE1 cleaving its substrates as a dimer or higher-order oligomer is well-established and compelling in the literature[1,6,7]. We do not think that G-1749 activates a monomeric form of IRE1: G-1749 engaged the IRE1 pathway uniquely in cells possessing pre-associated IRE1, i.e., KMS-11 (Fig. 6a). The co-crystal structure of G-1749 with IRE1 revealed a B2B dimer, with no major conformational changes in the RNase domain or in dimer symmetry, but showing a completely different conformation of the kinase activation loop (Fig. 2a, c). Our SAR analysis of this class confirms the need for an intact K599-

E612 salt bridge and an inward conformation of the C-helix, as previously shown for IRE1 allosteric activators, but it also reveals that for compounds like G-1749, optimal occupancy of a front pocket is required in order to achieve efficient activation (Fig. 3). Together, these results reveal that the conformation of the kinase activation loop is critical to the mechanism of RNase activation.

Precisely how the loop's conformation effects allosteric activation of the RNase remains unclear. In the oligomeric structure of *S. cerevisiae* IRE1 (Supplementary Fig. 9). The activation loop mediates a side-to-side interface, formed by B2B dimers stacking next to each other. Comparison of the crystal structure of G-1749 in complex with human IRE1 to a tetrameric unit of the *S. cerevisiae* IRE1 oligomer (Supplementary Fig. 9) reveals that the activation loop conformation induced by G-1749 is not compatible with the yeast IRE1 tetrameric interface. Perhaps this indicates that RNase allosteric activators from the G-1749 class are capable of selecting a different, more active, oligomeric conformation of IRE1. However, it is difficult to assess this notion since the oligomeric structure of human IRE1 has yet to be determined. In fact, human IRE1 could have a substantially different tetrameric interface than yeast IRE1, given the low sequence conservation of the activation loop between the two species. Alternatively, it is possible that the activation loop dynamics may allosterically control the conformation of the RNase, but only when IRE1 is associated in a B2B dimer. Our

HX-MS experiments, which were carried out at a final IRE1 concentration of 4 μM and in absence of RNA substrate, do not support a direct communication between the kinase and RNase domains of IRE1 within an isolated monomer: we did not observe any differential labeling at the kinase-RNase interface, nor in the RNase domain itself, in IRE1 samples containing activators G-1749 or G-9807, or inhibitor AMG-18, compared to the apo protein (Fig. 4d). At present, we also lack detailed understanding at the molecular level of RNA substrate binding to IRE1. Therefore, the true RNase active conformation of the protein remains elusive. Better definition of this could aid in SAR for allosteric activators as well as inhibitors. Further studies will be needed for full mechanistic elucidation of this class of IRE1 RNase allosteric activators.

An important feature of G-1749 is its shared chemical scaffold with the allosteric RNase inhibitor AMG-18. This allowed us to establish that allosteric inhibitors hinder IRE1 dimerization by occupying the back pocket of the ATP-binding site. While this mechanism was proposed before[20], our SAR studies on this scaffold show unequivocally how minor modifications can turn an allosteric activator into an inhibitor, solely by breaking the K599-E612 salt bridge and allowing the C-helix to be displaced. Furthermore, clear destabilization of the C-terminus of the C-helix in the presence of allosteric inhibitor AMG-18 is evident in our HX-MS studies (Fig. 4d).

We demonstrated that the ability of G-1749 and related compounds to allosterically activate the RNase is highly dependent on the phosphorylation state of IRE1: activators of IRE1-0P become inhibitors of IRE1-3P. This underlines the key role of the kinase activation loop conformation in the mechanism of action of G-1749. By producing a series of mutants to assess the role of each phosphorylation site, we observed that a single phosphorylation event on the IRE1 activation loop, regardless of the specific serine, does not appreciably increase RNase activity. However, two phosphorylation events are capable of significantly activating the RNase, with the combination of pS724 and pS726 being the most effective, as previously observed[8]. Correspondingly, G-1749 and its class of compounds activated IRE1-1P, while inhibiting IRE1-2P. More specifically, we found that R687, part of the catalytic HRD motif in the IRE1 kinase, is a central residue to this modulation switch: in absence of this sidechain within IRE1-3P, G-1749 class compounds can revert back to activation. This becomes even more pronounced upon substitution by alanine of both R687 and R611 (located on the C-helix). Both arginine residues interact directly with pS729, the most proximal phosphorylation site to the DFG motif, whose conformation must change dramatically to enable activation by the G-1749 class. Thus, when the R687-pS729 interaction is missing, G-1749 becomes agnostic to the presence of the three phosphorylation modifications, which otherwise would rigidify the loop in an extended configuration and make conformational changes by G-1749 energetically unfavorable.

The implication around the discovery of G-1749 and its unique modulation of IRE1 is that the RNase domain of IRE1 seems to be activated via the kinase domain in two ways. In one case, phosphorylations on the kinase activation loop can stabilize an extended conformation of the loop, rigidifying the kinase and stabilizing the B2B dimer as a consequence. The stabilization of the B2B dimer by phosphorylation can only be partially enhanced by ligands such as G-9807 or pan-kinase inhibitor Staurosporine[36], which only mildly activate IRE1-3P (Fig. 5a). This highlights how the mechanism of activation by phosphorylation and by small-molecule type I kinase inhibitors are potentially overlapping. In an alternative scenario, an upward conformation of the loop in absence of phosphorylation in a B2B dimer can increase activity of the RNase domain. This kind of conformation

has been observed before in two separate crystal structures of IRE1 in a F2F dimer conformation (PDB: 3P23 https://doi.org/10.2210/pdb3P23/pdb, 4YZD https://doi.org/10.2210/pdb4YZD/pdb)[33,36], in both cases in complex with ADP, which has been reported to activate the RNase domain allosterically[37,38]. We show in our work that this kind of activation can be brought upon by small molecules such as G-1749 and G-7658, but whether this modulation is physiologically relevant remains to be seen.

In conclusion, our work diversifies the available collection of IRE1 modulators by adding a class of selective IRE1 allosteric activators while providing insight into the nature of kinase-RNase interdomain regulation within IRE1. These studies open a distinct avenue for exploring the potential therapeutic benefits of IRE1 activators in disease.

## Methods

**IRE1 RNase activity assay**. A 5′-Carboxyfluorescein (FAM)-and 3′-Black Hole Quencher (BHQ)-labeled single stem-loop mini-substrate containing XBP1 sequence (sequence reported in Supplementary Table 4) was used as substrate for cleavage by IRE1 LKR (Q470-L977) or IRE1 KR (G547-L977) in 20 mM HEPES pH 7.5, 50 mM potassium acetate, 1 mM magnesium acetate, 1 mM dithiothreitol, 0.05% v/v TritonX-100. Concentrations were as follows: 8 nM IRE1 LKR and 200 nM RNA substrate; 22 nM IRE1 KR-0P and 200 nM RNA substrate; 2 nM IRE1 KR 3P and 500 nM RNA substrate. RNA cleavage was measured kinetically over an hour at room temperature as an increase in fluorescence. The final reaction was carried out in 20 μL in 384-well plates. Samples were run in duplicate.

**Branched DNA (bDNA) assay**. 15-20k KMS-11 cells in RPMI growth media were seeded into 96-well plates and treated with compounds or DMSO control over a period of 4 h the following day. RNA levels of XBP1s and DGAT2 in KMS-11 cell lysates were measured using the QuantiGene 2.0 bDNA assay technology (Thermofisher). See Supplementary Table 4 for sequences of relevant capture probes. Data analysis was carried out against an assay background control with no cell lysate (no-RNA control). In the case of AMG-18, cells were also stimulated with thapsigargin (Tg, 100 nM) 30 min after compound addition. No Tg was used when treating cells with all other compounds. Samples were run in duplicate.

**Generation of MDA-BD-231 IRE1 KO cell line by CRISPR/Cas9 knockout: guide RNA sequences and technique**. MDAMB231 IRE1 KO cells were generated using CRISPR by co-transfecting a Cas9 containing plasmid, pRK-TK-Neo-Cas9, with a pair of IRE1 targeting gRNAs (see Supplementary Table 4 for sequences) cloned into a pLKO vector. Transfection was done using Lipofectamine 3000 (Invitrogen #L3000008) according to the manufacturer's protocol. Transformants were selected by PCR on genomic DNA for the detection of deletions. Correct clones were then sequenced.

**Cell treatment with compounds and western blots of crosslinked lysates**. KMS-11 and MDA-MB-231 cells were obtained from ATCC, authenticated by short tandem repeat profiles, and tested to ensure they were mycoplasma-free within 3 months of use. All cell lines were cultured in RPMI1640 media supplemented with 10% (v/v) FBS (Sigma), 2 mM glutaMAX (Gibco), and 100 U/mL penicillin plus 100 μg/mL streptomycin (Gibco). Thapsigargin (Sigma) was used at a concentration of 100 nM. Compounds were dissolved as 10 mM stocks in DMSO and used at the indicated final concentrations, maintaining a constant final concentration of 0.06% DMSO. Antibody for GAPDH (5174) was from Cell Signaling Technology. Antibodies for detection of IRE1 and XBP1s were generated at Genentech. Secondary antibody (711-035-152) was from The Jackson Laboratory. IRE1 kinase-dead mutant D688N cDNA used in transfection experiments was cloned in a pRK-TK-Neo vector driven by a CMV promoter.

For IRE1 transfection experiments, MDA-MB-231 IRE1 KO cells were grown to 70–90% confluence in 10 cm dishes (Corning) and then transfected with 20 μg of IRE1 DNA (or water for control condition) using the Lipofectamine LTX with Plus Reagent DNA transfection protocol (Thermofisher). Cells were incubated for about 48 h, then the same protocol as the one used for standard serial dilution experiments was followed. For serial dilution experiments, cells were seeded in standard 6-well plates (Corning) at $1 \times 10^6$ cells per well and incubated overnight at 37 °C. Compounds were incubated with the cells for 4 h. Cells were lysed with 1% Triton X-100 in PBS supplemented with protease and phosphatase inhibitors (Thermofisher Scientific #78440). Lysates were spun at $15,000 \times g$ for 10 min. Clear lysates were incubated with a solution of 0.25 mM DSS in DMSO (Thermofisher) for 1 h at room temperature and 500 rpm shaking. Crosslinked samples were quenched with 50 mM TRIS pH 7.5 for 15 min at room temperature and 500 rpm shaking. Protein concentration was measured by BCA assay (Thermofisher Scientific #23227) and 25 μg of protein lysate was loaded on a 4-12% Bis-Tris NuPage gels and run in MOPS buffer (Thermofisher Scientific # NP0321 and #

NP000102) for 3 h at 100 V to achieve optimal resolution of crosslinked IRE1. Proteins were transferred onto a PVDF membrane using iBlot 2 (Thermofisher Scientific # IB24001). Membranes were blocked in 5% milk at room temperature for 30 min to an hour, then supplemented with primary antibodies (1:1000): IRE1α Lumenal Domain (LD) (mouse monoclonal, IgG2a) and XBP1s (rabbit monoclonal) were both generated at Genentech[15,39], GAPDH (5174) was from Cell Signaling Technology. The membranes were then incubated overnight at 4 °C. Membranes were washed three times (5 min each) with PBST buffer, then incubated in 5% milk supplemented with secondary antibody (1:10000, Peroxidase AffiniPure Donkey Anti-Rabbit IgG, 711-035-152, The Jackson Laboratory) and incubated at room temperature for 1–3 h. After washing (3 × 5 min) with PBST buffer, membranes were immersed in SuperSignal West Dura or Femto substrate (Thermofisher Scientific # 34075 × 4 and # 34096 × 4) for 1-5 min, patted dry, and imaged using Azure 600 Imager (Azure Biosystems).

**Sedimentation velocity analytical ultracentrifugation (SV-AUC)**. IRE1 LKR (Q470-L977) at concentrations of 3.3, 10, and 30 μM in 25 mM HEPES pH 7.5, 250 mM NaCl, 1 mM TCEP was incubated with a molar excess of compound in DMSO (4, 12, and 35 μM, respectively; final %DMSO constant) and incubated for 1 h at ambient temperature. Sedimentation velocity experiments were run at 20 °C and 45,000 rpm. Sedimentation was monitored by $A_{280}$. 140 scans were collected for each run. Apparent sedimentation coefficient (s*), g(s*) distribution, and fit to a monomer-dimer equilibrium model were calculated using Sedanal software[40,41].

**Protein purification and separation of phosphorylated IRE1 fractions**. IRE1 KR (G547-L977) or IRE1 LKR (Q470-L977) were expressed as N-terminal His$_6$-tagged fusion proteins in Sf9 cells with a TEV protease cleavage site from an intracellular BEVS expression vector. Cell pellet was resuspended in lysis buffer containing 50 mM HEPES pH 8.0, 300 mM NaCl, 10% glycerol, 1 mM MgCl$_2$, 1:1000 benzonase, EDTA-free PI tablets (Roche), 1 mM TCEP, and 5 mM imidazole. Sample was lysed by sonication, centrifuged at 15,000 × g for 45 min, and the supernatant filtered through a 0.8 μ Nalgene filter. Cleared supernatant was bound to Ni-NTA Superflow beads (Qiagen) by gravity filtration. Beads were washed in lysis buffer supplemented with 15 mM imidazole, followed by protein elution in lysis buffer containing 300 mM imidazole. The eluate was incubated with TEV protease overnight at 4 °C. The sample of IRE1 KR protein was diluted 1:10 in 50 mM HEPES pH 7.5, 50 mM NaCl, 1 mM TCEP and then loaded onto a 5 mL pre-packed Q-HP column (GE Healthcare). Separation of IRE1 KR unphosphorylated and phosphorylated was achieved by eluting the protein with a very shallow gradient (50–300 mM NaCl over 70 CV). Fully phosphorylated fraction (MW + 240 by LC-MS, see Supplementary Fig. 11) was collected separately, while the rest of the protein fractions were consolidated and incubated with Lambda phosphatase for 1 h at room temperature. Dephosphorylation was confirmed by LC-MS (see Supplementary Fig. 11). Unphosphorylated and phosphorylated samples were then concentrated and loaded separately onto a HiLoad 16/600 Superdex 200 SEC column (GE Healthcare) equilibrated in 25 mM HEPES pH 7.5, 250 mM NaCl, 1 mM TCEP, 10% glycerol. IRE1 eluted as a monomer. The purification for IRE1 LKR differed in the following details: IRE1 LKR was incubated overnight with both TEV protease and Lambda phosphatase after elution from Ni-NTA beads. The resulting unphosphorylated sample was purified by SEC and eluted as a monomeric peak. Phosphorylated LKR for SV-AUC was generated according to the protocol described separately in the Methods. Correct location of phosphorylation sites was confirmed by LC-MS/MS (see methods for details) and is reported in Supplementary Table 3.

**Phosphorylation of IRE1 LKR and activation loop mutants**. IRE1 LKR was allowed to autophosphorylate in the presence of 2 mM ATP and 10 mM MgCl$_2$ for 1 h at room temperature. A distribution of three to eleven phosphorylations was observed by LC-MS. Sample was purified from ADP by size-exclusion chromatography (SEC).

IRE1 KR S/A and X/A activation loop mutants were also allowed to autophosphorylate by the same method. Mutants S726A/S729A, R611A/R687A/ K716A, and R611A/R687A/K716A/R722A/N750A were not capable to autophosphorylate and were instead incubated with pIRE1 LKR at 1:40 w/w, 20 mM ATP and 20 mM MgCl$_2$. Final phosphorylated proteins were purified from pIRE1 LKR and residual nucleotides by SEC.

Phosphorylation sites for all proteins were identified by LC-MS/MS (see methods for details) and are reported in Supplementary Table 3.

**LC-MS/MS and phosphorylation site mapping**. Twenty μg of each protein sample were reduced with 20 mM DTT at 37 °C for 30 min followed by alkylation with 40 mM iodoacetamide at room temperature for 30 min. Samples were split in half for separate enzymatic digestion with 0.1 μg trypsin (Promega) and 0.1 μg chymotrypsin (Thermo Fisher Scientific) at 37 °C overnight. Digests were quenched with 0.1% TFA and subjected to C18 stage-tip clean up with a 50% acetonitrile containing 49.9% water plus 0.1% TFA elution step. After clean up, peptides were dried down and reconstituted in 50 μl 0.1% TFA, where 1 μl was injected onto a UPLC system (Waters NanoAcquity) via an autosampler and separated on a 45 °C heated Acquity M-Class BEH C18 column (0.1 mm ×

100 mm, 1.7 μm resin, Waters). A binary gradient pump was used to deliver solvent A (97.9% water, 2% acetonitrile and 0.1% formic acid) and solvent B (97.9% acetonitrile, 2% water and 0.1% formic acid) as a gradient of 2% to 25% solvent B over 35 min at 1 μl/min. The solvent was step-changed to 50% solvent B over 2 min and then held at 90% for 6 min to clean the column. Finally, the solvent was step-changed to 2% solvent B and held for 7 min for re-equilibration. Separated peptides were analyzed on-line via nanospray ionization into an Orbitrap Elite Hybrid Ion Trap-Orbitrap mass spectrometer (Thermo Fisher Scientific) using the following parameters for data acquisition: 60,000 resolution; 375-1,600 m/z scan range; positive polarity; centroid mode; 1 m/z isolation width with 0.25 activation Q and 10 ms activation time; CID activation; and a CE setting of 35. Data were collected in data-dependent mode with the precursor ions being analyzed in the FTMS and the top 15 most abundant ions being selected for fragmentation and analysis in the ITMS. Acquired mass spectral data were searched against the protein sequence using Byonic software (Protein Metrics Inc.) with the following parameters: 20 ppm precursor mass tolerance with 0.5 Da fragment mass tolerance; strict specificity on arginine and lysine with up to 1 missed cleavage for trypsin; strict specificity on leucine, phenylalanine, tryptophan, and tyrosine with up to 2 missed cleavages for chymotrypsin; fixed carbamidomethylation on cysteine; variable oxidation on methionine; and variable phosphorylation on serine, threonine, and tyrosine. Byonic search results were analyzed in the Byologic software (Protein Metrics Inc.) and filtered with a minimum MS2 score cutoff of 200. Peptide identification was confirmed by MS2 peptide fragmentation. Phosphorylated peptides were label-free quantified relative to its unmodified form by AUC integration of their extracted ion chromatograms in Thermo Xcalibur Qual Browser software (Thermo Fisher Scientific). The full results of the phosphorylation mapping experiments are reported in Supplementary Data 5.

**X-ray crystallography**. Protein crystals were grown as reported below. All data were indexed and reduced using the autoPROC package (Global Phasing Ltd.)[42] (with the exception of the cpd. 2/IRE1-3P and G-9807/IRE1-0P datasets, which were processed using iMOSFLM[43]). Some of the crystals exhibited anisotropic diffraction, therefore anisotropic scaling was performed on those datasets with STARANISO software[44]. Data reduction statistics are reported in Table 1, which includes resolution estimates derived from both isotropic and anisotropic scaling. The structure was phased by molecular replacement using Phaser (CCP4)[45], and the STARANISO maps were used for model building in Coot[46] and refinement using Phenix[47].

**G-9807 - IRE1-0P co-crystallization**. The crystals of G-9807 in complex with IRE1 0P were obtained by Proteros. The protein in 20 mM Tris/HCl pH 8, 150 mM NaCl, 2 mM DTT was used at 9 mg/ml. It was incubated for 2 h with 2 mM of the ligand (diluted from a 100 mM DMSO stock).

The reservoir solution was 10% (v/v) isopropanol, 11% (w/v) PEG4000, 0.1 M Na-citrate, pH 5.6. Crystallization was done in 24 well Linbro plates, hanging drop. 0.5 μl + 0.5 μl were equilibrated over 0.5 mL reservoir at 4 °C. Crystals appeared after 2 days and grew to the final size in 4 days. 25% glycerol in reservoir was used as cryoprotectant before flash freezing in liquid nitrogen. Data collection was done at the Swiss Light Source (SLS, Villigen, Switzerland).

**G-1749 - IRE1-0P co-crystallization**. IRE1 KR 0P at a concentration of 16 mg/mL was mixed with a DMSO solution of G-1749 (1 mM final concentration), incubated for 1 h in ice, then spun at 14,000 rcf for 15 min. Thin plates grew over 2–3 days in sitting drop-vapor diffusion (SDVD) set up with TTP Labtech Mosquito with mother liquor containing 0.1 M MES pH 6.5, 0.2 M magnesium acetate, 18% w/v PEG 8 K at 4 °C. Crystals were cryoprotected in the mother liquor supplemented with 25% glycerol and flash frozen in liquid nitrogen. Data collection was done at the Advanced Light Source, beamline 5.0.2. (ALS, Berkeley, CA, U.S.A).

**G-7658 - IRE1-0P co-crystallization**. IRE1 KR 0P at a concentration of 14 mg/mL was mixed with a DMSO solution of G-7658 (1 mM final concentration), incubated for 1 h on ice, then spun at 14,000 rcf for 15 min. Thin plates grew over a week in SDVD set up with TTP Labtech Mosquito with mother liquor containing 0.2 M magnesium formate, 20 %w/v PEG 3350, and 1% CYMAL-1 as additive, at 4 °C. Crystals were cryoprotected in the mother liquor supplemented with 35% PEG400 and flash frozen in liquid nitrogen. Data collection were done at the Stanford Synchrotron Radiation Light source, beamiline 12-2 (SSRL, Stanford, CA, U.S.A).

**Compound 2 - IRE1-3P co-crystallization**. IRE1 KR 3P at a concentration of 15 mg/mL was mixed with a DMSO solution of Compound **2** (1 mM final concentration), incubated for 1 h on ice, then spun at 14,000 rcf for 15 min. Small rods grew over two-three days in SDVD set up with TTP Labtech Mosquito with mother liquor containing 0.1 M HEPES pH 7.5, 0.1 M potassium dihydrogen phosphate, 16 %w/v PEG 8K, and 5.2 mM FOS-MEA-10 as additive, at 4 °C. Crystals were cryoprotected in the mother liquor supplemented with 25% glycerol and flash frozen in liquid nitrogen. Data collection was done at the Stanford Synchrotron Radiation Lightsource, beamiline 12-2 (SSRL, Stanford, CA, U.S.A).

**IRE1-3P apo crystallization**. IRE1 KR 3P at a concentration of 12 mg/mL was incubated with DMSO (3% final concentration), incubated for 1 h in ice, then spun at 14,000 rcf for 15 min. Small prisms grew after two-three days over about a week in sitting drops set up with TTP Lab Tech Mosquito with mother liquor containing 10% 2-propanol, 0.1 M sodium citrate tribasic dihydrate pH 5.0, and 26% PEG400, and 4% pentaerythritol ethoxylate, 4% 1,3-propanediol, 4% 1,3-butanediol as additives, at 4 °C. Crystals were flash frozen in liquid nitrogen. Data collection was done at the Canadian Light Source, beamline 08ID, (CLS, Saskatoon, Canada).

**IRE1-0P apo crystallization**. IRE1 KR 0P at a concentration of 12 mg/mL was incubated with DMSO (3% final concentration), incubated for 1 h on ice, then spun at 14,000 rcf for 15 min. Rods grew after 3–4 days over about ten days in SDVD set up with TTP Lab Tech Mosquito with mother liquor containing 0.1 M Hepes pH 7.5, 42% PEG200, and 1 M lithium chloride, 3% w/v 6-aminohexanoic acid as additives, at 4 °C. Crystals were flash frozen in liquid nitrogen. Data collection was done at the Canadian Light Source, beamline 08ID, (CLS, Saskatoon, Canada).

**Hydrogen-Deuterium Exchange (HX-MS)**. IRE1 KR was exchanged to HX-MS buffer (10 mM Histidine pD$_{read}$ 7.0, 250 mM NaCl, 1 mM TCEP) using Zeba spin desalting columns (Thermofisher) and diluted to a final concentration of 45 µM. Compounds (or the same volume of DMSO) were added to the protein solution as DMSO stocks to a final concentration of 70 µM and incubated for about 1 h on ice before initiating exchange as follows. 4 µL of sample was mixed with 60 µL of a matched deuterated HX-MS buffer and incubated for 0.5, 2.6, 12, 58, 166, or 480 min at 5 °C. The labeling was quenched by addition of 55 µL of quench buffer (4 M guanidine, 400 mM glycine, 0.25 M TCEP, final pH = 2.3) to 55 µL of labeled protein sample. 100 µL of quenched protein sample were injected onto a pepsin:protease XIII 50:50 column (NovaBioassays) and the mixture of peptides were then separated by a BEH C18 HPLC column (Waters Acquity UPLC) and eluted in a water/acetonitrile gradient. MS measurements were made using a ThermoScientific Q-Exactive HF-x instrument where peptide fragmentation occurred by HCD and in excess of 350 unique peptides were tracked throughout all experiments described herein providing greater than 96% sequence coverage. Our experimental manipulations were automated using a custom-built Leap HDX Pal DHR platform by Leap Technologies, Morrisville, NC. Each sample was run in duplicate.

An MS/MS experiment, used to define the retention times and identities of peptides tracked during the labeling experiments, was collected in a data-dependent top ten mode with a max fill time of 110 ms, 60k HZ resolution of parent scans, 7.5k Hz resolution on fragment spectra, and deuterated spectra were collected at 120k Hz in ms-only mode with a max fill time of 200 ms. Peptides were identified using Byonic (version 3.2, Protein Metrics Inc.) with precursor and fragment mass tolerance of 6 ppm, and further filtered by Byologic (version 3.2, Protein-Metrics Inc.) using a PEP2D cutoff of >0.001. Raw files from the mass spectrometer were converted into an open-source format (mzXML) using the publicly available MSconvert GUI[48] (version 3.0.19256-a8cbe7417) and extracted ion chromatograms of deuterated peptides were produced using the ExMS program version 1[49]. Peptides were charge-state-averaged, and deuterium uptake information extracted using algorithms described exactly in chapter 4.4 and appendix A[50]. Changes in protection factors were extracted using an empirical algorithm[35]. In a manner largely consistent with standards set forward recently for the dissemination of HX-MS data[51], we have included plots of every peptide measured and used in this work (see Supplementary Data 2) and listed the computed peptide protection factors individually for each condition tested along with other useful experimental details (See Supplementary Data 3). We have not included a list of deuterium uptake values for each time point for every peptide included in this work because the peptide protection factor with range of observations given in Supplementary Data 3 reduce this information into a more useful format. Additional information can also be found in Supplementary Data 4. Note: PDB files use residues numbers that are substantially offset from the sequential numbering used in the peptide plots and tables. For reference, residue 16 in HX data is equivalent to residue 562 in the PDB files referenced herein (see also Supplementary Fig. 1a).

**Reporting summary**. Further information on research design is available in the Nature Research Reporting Summary linked to this article.

## Data availability

Coordinates and structure factors of structures reported in this manuscript have been deposited in the Protein Data Bank with the following accession codes: 6W39, 6W3A, 6W3B, 6W3C, 6W3E, 6W3K. The authors declare that all other data supporting the findings of this study are available within the article and its Supplementary Data files, or from the corresponding authors on request. Source data are provided with this paper.

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

## Acknowledgements

This paper is dedicated to the memory of our late co-author Susan E. Kaufman. We thank the members of the laboratory of Prof. Peter Walter (University of California San Francisco) and Avi Ashkenazi (Genentech) for helpful scientific discussion and support. We thank the members of the following departments at Genentech for technical support: Biomolecular Resources, Discovery Chemistry (Purification and Analytical groups), Structural Biology, Microchemistry, Proteomics & Lipidomics, Biochemical and Cellular Pharmacology. We thank the staff of Advanced Light Source, Stanford Synchrotron Radiation Lightsource, Swiss Light Source, and Canadian Light Source for provision of synchrotron facilities and for assistance during data collection.

## Author contributions

E.F., W.W., J.R., A.A. conceived and designed the study. E.F. performed the majority of experiments and analyzed data. A.LeT. generated the CRISPR KO cell line, and provided support with cell experiments. H.A.W and W.W. performed crytallography for IRE1 apo 3P and 0P. A.L. performed crytallography for IRE1 in complex with G-9807. E.S.D. provided support for SV-AUC experiments, analyzed data, and wrote the SV-AUC section of the manuscript. B.T.W. provided support for HX-MS experiments, analyzed data, and wrote the HX-MS section of the manuscript. S.E.K. provided support for the bDNA assays. M.-G.B. directed the synthesis of G-1749. K.R.C. provided support with the RNase activity assay. M.H.B. provided support with all biochemical and cellular assays. K.M. performed the expression of IRE1 protein. Y.-C.A.C. provided support with cell lysates crosslinking experiments. B.C. synthesized all the compounds reported in Fig. 3c. W.P and P.S.L performed the phosphorylation mapping experiments. E.F., W.W., J.R., A.A. wrote the manuscript. All authors discussed results and commented on the manuscript.

## Competing interests

E.F., A.LeT., H.A.W., E.S.D., B.T.W., S.E.K., M.G.B., K.R.C., M.H.B., K.M., Y.A.C., B.C., W.P, P.S.L, A.A., J.R., W.W. were employees of Genentech, a member of the Roche Group, while working on this project. A.L. is an employee of Proteros Biostructures.
