## [Peer Review File · Nature Communications]

REVIEWER COMMENTS

Reviewer #1 (Remarks to the Author):

The manuscript by Ferri et al describes the activation/inhibition of RNase activity of IRE1 through small molecule ligands that bind in the ATP binding pocket of the kinase domain. IRE1 is a key protein in the unfolded protein response. The authors describe the identification of a novel G-1749 compound that activates IRE1 through a previously unknown conformational driven mechanism, distinct from the formation of a B2B dimer. Intriguingly this activates pre-phosphorylated IRE1, but inhibits the phosphorylated form. This was well characterised using a number of methodologies, including HDX-MS and X-ray crystallography.

Overall, this is an exciting high quality manuscript, using a variety of well controlled and carried out biophysical methods. There are detailed structure activity relationships that define how different molecules can either activate or inhibit RNase activity of IRE1. The development of these compounds could act as potent tools to understand the unfolded protein response. This manuscript should be suitable for publication after only minor changes in text and figures. Here are the following major and minor points I found that could improve this manuscript.

Major points

1. The HDX-MS data shown in Fig 4b is hard to interpret as currently shown. The error bars are hard to make out, and the lines are so thick, that it makes analysis challenging. I am not sure they need to show the intrinsic rate of this sequence in the main text figure, as this is shown in the supplement, as in this view they are minimising the actual changes they see upon the different conditions.
2. Could the decreased protection factor shown for the DLKPHNIL peptide seen for G-9807 also possibly be explained by an altered affinity of the B2B dimer conformation? HDX was carried out at 4 μ M, what is the calculated fraction in a dimer at this concentration?

Minor points.

1. In figure 1 when comparing in vitro RNase activity vs XBP in cells (Fig 1b vs 1c) there is a much less pronounced effect of G-9807 on in vitro RNase activity compared to G-1749, yet the cellular effects are similar. Is this driven through the difference in oligomerisation between the two (Fig 1e)? Do the authors have a specific hypothesis for this?

2. The crystallisation statistics show some strange trends in how the resolution cutoff was selected.

For example the IRE1-OP/G-9807 structure. Here the $I/\sigma I$ is relatively high in the final shell ~ 2.5 , with a high $CC1/2$ (0.84). Did the authors try extending the data to higher resolution for some of these structures to see if this improved the electron density?

Could the authors also include Fo-Fc density maps for their inhibitors as a supplement?

3. Could the authors highlight the locations of the phosphorylation sites in the Fig 2 structure as this will ease interpretation of later figures.

4. It is particularly striking how well the authors have defined activation vs inhibition with the different compounds shown in Fig. 3. This is particularly evident for the comparison of G-1749 vs compound 7, where a single methyl group in the tail completely changes the effect on RNase activity.

The authors highlight this tail region in panel 3d. However this overlay makes it very hard to interpret the hydrogen bonds, and in addition there currently is no figure legend for this panel, making it hard to understand.

5. It is very hard to interpret the Fig S8 conserved residue figure. Is there a better way to show this rather than the disjointed surface representation?

6. For many of the HDX plots in the supplemental data there are curves that lack some time points. Was this do overlapping peptides at longer D2O exposures?

Author report- John E Burke

Reviewer #2 (Remarks to the Author):

The paper by Ferri et al describes the effects of ATP competitive inhibitors on IRE1 activity and suggests that the kinase loop conformation plays an important role in modulating activity. They identify an inhibitor G1749 that displays unusual behaviour which they assess using structural, in vitro and cellular approaches.

Overall, the paper is reasonably interesting and attempts to explain contrasting IRE1 activity on kinase loop conformations. Although this does not explain how this modulates RNase splicing, it does describe an important regulatory aspect of IRE1 activity and therefore warrants publication. It is already known that kinase loop phosphorylation greatly enhances splicing. The technical approach is fairly well thought out and executed well enough but for one major addition/clarification required see below. The conclusion that are drawn are in line with the data presented in the paper.

Major point

In figure 6 the data shows that there is an increase in splicing in KMS-11 cells upon addition of G1749. However, one might expect that in these cells G1749 would inhibit splicing since the IRE1 pathway is constitutively activated, suggesting a higher population of phosphorylated IRE1. G1749 was shown convincingly to inhibit splicing of phosphorylated IRE1 in the earlier experiments. Similarly, in cells that require chemical induction of UPR one would expect that addition of G1749 would activate splicing as there would be low levels of phosphorylated IRE1, particularly before the addition of Tg. This is further supported by the levels of IRE1 dimer when crosslinked. There is a higher proportion of dimers and thus higher splicing in the constitutively activated KMS-11 cell line. Thus, the results obtained using G1749 are the opposite of what one might expect. The key missing information is the direct measure of phosphorylation in these experiments. It is important to assess the levels of IRE1 phosphorylation either by using a specific phospho antibody (p724 antibody) or a phos-tag or similar.

Minor points (not required to do).

In the SV experiments the concentration is not necessarily shown as molar excess of inhibitor, but a 1:1 ratio which is fine but then they should state this.

It would be nice to see a secondary measure of monomer-dimer IRE1 bound to inhibitor using cross linking in Figure1 and throughout.

If the data is available, is the salt-bridge K599-E612 intact in G1749-IRE13P structure?

It would be nice to see a mutation that disrupts the above-mentioned salt bridge and then observe its effects on splicing and dimer formation in solution.

It would be nice to see a mutation that disrupts the b2b dimer based on the structure and then to measure if this then inhibits dimer in solution or not.

Do the cellular crosslinking experiments show oligomer formation such as tetramer or larger?

Reviewer #3 (Remarks to the Author):

This manuscript presented by Ferri et. al., features the development of IRE1-selective kinase inhibitor-RNase activators. The authors have identified two novel allosteric activators, G-9807 and G-1749, of which G-1749 displays high kinase-selectivity. The mechanism of activation for these novel activators was rationalized with co-crystal structures of IRE1, HD-MX, and both biochemical and cellular RNase activity assays. From these experiments the authors concluded that G-9807, typical of type I kinase inhibitors, favors IRE1 B2B dimer formation and consequently RNase activity. Conversely, G-1749 accesses a region they call the front pocket, which stabilizes IRE1's activation loop in a distinct conformation. As a consequence, G-1749 inhibits phosphorylated IRE1 (IRE1-3P), but activates dephosphorylated IRE1 (IRE1-0P). G-1749 activates IRE1 through an unknown mechanism as no B2B dimers were observed. This work culminates with a cellular experiment, where the authors show that G-1749 is only capable of activating pre-associated IRE1. This study presents a comprehensive analysis of IRE1 kinase inhibitor/RNase activators, incorporating crystallography and HD-MX alongside biochemical and cellular characterization. The molecules presented in this work are a nice addition to the chemical tools that are currently available for IRE1 and the UPR. Specifically, the generation of IRE1-specific allosteric activators are the first of their kind and will be an essential tool to understanding IRE1's unique role as part of the UPR. While the

manuscript is well-written, the authors will need to address some concerns below before the manuscript can be accepted for publication.

1. The concentrations required for SV-AUC experimentation are incredibly high. The lowest concentration of IRE1 used in these assays is 3.3 μ M. However, although the concentration of IRE1 used in the RNase assays is not included in the figure legends or experimental (which should also be addressed), it can be assumed that these assays were ran under Michaelis-Menten kinetics where $[IRE1] \ll [RNA]$ used. This would suggest that concentrations of IRE1 used in the RNase assay to obtain robust XBP1 cleavage are much lower than those used in SV-AUC experiments (see Figure S6). How do the authors justify measuring Kdimer at such high concentrations, while also using much lower concentrations in their RNase assays (assuming dimerization is necessary for activity in this assay)?
2. A major claim of the paper is that G-1749 does not stabilize dimers, contrary to previously published work on allosteric activators. This would fundamentally change how we think about the kinase-RNase allosteric network. Considering this, more needs to be done to convince the reader that G-1749 does not stabilize dimers. This claim could be bolstered by performing additional chemical crosslinking experiments or FRET-based dimerization experiments. Alternatively, the argument that G-1749 does not stabilize dimerization would be much more convincing if the authors displayed activation of IRE1 with G-1749 using IRE1-OP with a mutant in the B2B dimer interface (e.g. D620). In general, I think the authors current evidence claiming G-1749 functions to activate IRE1 independently of dimerization needs stronger biochemical evidence.
3. The SAR analysis of front pocket binding is puzzling. The authors state the front pocket has a limited tolerance for chemical substituents; stating that bulkier substituents cannot occupy the front pocket. But inhibitors that have smaller substituents (comp 4 and 5), that presumably could fit into the front pocket are shown to inhibit the RNase domain. Considering that all allosteric RNase inhibitors tested in this focused SAR lack groups that can accept electrons from the carboxylate of Glu612 (such as a carbamate or acrylamide), couldn't this instead suggest that to be an activator, there needs to be an electron accepting group to coordinate Glu612 of the α C-helix anchoring it an inward position?
4. Table S3 should include SEM values.
5. In Figure 6 the authors use DSS, an amine-reactive chemical crosslinker, to chemically crosslink IRE1. This is problematic as AMG-18 and G-1749 both have free-amines that can react with DSS. Considering the stoichiometric excess of small molecule to endogenous IRE1 wouldn't one expect a majority of the DSS to be quenched by inhibitor?
6. Authors need to show that G-1749 and G-9807 are engaging IRE1 to a similar extent in cells. The results shown could be due to different levels of engagement in the respective cells lines. Additionally, because phosphorylation state is important to the mechanism of G-1749, the authors should also characterize the phosphorylation status of IRE1 in cells using phostag gels.

7. Can the authors speculate further on why G-1749 would only activate pre-associated IRE1? Isn't this contrary to what is observed in vitro, where G-1749 can activate IRE1-OP using concentrations of IRE1-OP that are well below 65 μ M (the reported KDIMER for IRE1-OP)?

Changes to Figures in the Manuscript

Figure 1

- Panel b was modified: while completing the Source Data file, we realized that incorrect data for AMG-18 had been included in Panel b. The figure was modified with the correct data.

Figure 3

- Panel d lacked a description – we added a legend for it.
- Panel d was modified to address Reviewer's #1 concerns.

Figure S4

- Figure S4 was modified to reflect the final crystal structure of the G-1749/IRE1 complex that was deposited in the Protein Database (PDB). The updated model for this crystal structure truncates the visible activation loop at T734 instead of E735 as previously shown in the figure. Additionally, ten new models were calculated for the missing loops using the new structure as a reference in the MOE Homology Model algorithm. The model reported in orange in this Figure was updated in Figure 4a.

Figure 4

- Panel a was updated using a new model that was calculated using the final crystal structure of the G-1749/IRE1 complex that was deposited in the PDB. See modifications to Figure S4.
- Panel b was modified according to Reviewer #1's comments.
- Panel c was modified by adding underlying data points to the bar graph, as outlined in the instructions to the authors.
- Panel d was modified to reflect changes made to the script that allowed to visualize the calculated protection factors (pF) on the crystal structures: the script used to produce the older panel had a mistake that did not assign the pF of the shortest peptide to all residues for which it should have done so.

Figure 5

- Panel f was modified by adding underlying data points to the bar graph, as outlined in the instructions to the authors.

Figure S8

- We have updated the Figure with a new representation where only the C α atoms of the conserved residues are highlighted as spheres, along a cartoon representation of the monomer of IRE1, in response to a comment of Reviewer #1. We have also updated the legend accordingly.

Figure S10

The Figure was added in response to Reviewer #1's request.

Responses to reviewers

(reviewers' comments are quoted in blue and italicized)

Reviewer #1

The manuscript by Ferri et al describes the activation/inhibition of RNase activity of IRE1 through small molecule ligands that bind in the ATP binding pocket of the kinase domain. IRE1 is a key protein in the unfolded protein response. The authors describe the identification of a novel G-1749 compound that activates IRE1 through a previously unknown conformational driven mechanism, distinct from the formation of a B2B dimer. Intriguingly this activates pre-phosphorylated IRE1, but inhibits the phosphorylated form. This was well characterised using a number of methodologies, including HDX-MS and X-ray crystallography.

Overall, this is an exciting high quality manuscript, using a variety of well controlled and carried out biophysical methods. There are detailed structure activity relationships that define how different molecules can either activate or inhibit RNase activity of IRE1. The development of these compounds could act as potent tools to understand the unfolded protein response. This manuscript should be suitable for publication after only minor changes in text and figures. Here are the following major and minor points I found that could improve this manuscript.

We sincerely thank the Reviewer for the positive remarks and constructive suggestions regarding our manuscript and approaches. Please see below for responses to the Reviewer's specific questions.

Major points

1. The HDX-MS data shown in Fig 4b is hard to interpret as currently shown. The error bars are hard to make out, and the lines are so thick, that it makes analysis challenging. I am not sure they need to show the intrinsic rate of this sequence in the main text figure, as this is shown in the supplement, as in this view they are minimising the actual changes they see upon the different conditions.

We have graphed the data without the intrinsic rate of the sequence, modified the Y axis scale to show the rates in full, and thinned out the lines. This has also rendered the zoomed inset unnecessary. We believe these changes have made the figure more informative and we thank the Reviewer for the suggestions.

2. Could the decreased protection factor shown for the DLKPHNIL peptide seen for G-9807 also possibly be explained by an altered affinity of the B2B dimer conformation? HDX was carried out at 4 μ M, what is the calculated fraction in a dimer at this concentration?

The DLKPHNIL peptide did not show any difference in deuteration between the samples. This result, as the Reviewer correctly pointed out, leads to a decreased

protection factor for G-9807 for this peptide compared to the peptide reported in Figure 4b. We think that the comparison between these two deuteration rates highlights residue L689 as the one being differentially deuterated in this region of the protein depending on the sample. This residue (and the kinase catalytic loop in general) is not involved in the B2B dimer interface, and, as such, we do not expect differences in this region to inform stability of the B2B dimer in the presence or absence of phosphorylations or small molecules. IRE1 was shown to be monomeric at 4 μ M in SV-AUC experiments (see lines 557-560 in the updated main text), so we expect the amount of stable IRE1 dimers at this concentration to be extremely low, which also would make it difficult to infer B2B dimer stability differences in the samples.

Minor points.

1. In figure 1 when comparing in vitro RNase activity vs XBP in cells (Fig 1b vs 1c) there is a much less pronounced effect of G-9807 on in vitro RNase activity compared to G-1749, yet the cellular effects are similar. Is this driven through the difference in oligomerisation between the two (Fig 1e)? Do the authors have a specific hypothesis for this?

When comparing the activity of the allosteric activators in the biochemical RNase assay (Figure 1b), G-9807 increased the activity of the IRE1 RNase about 1.5-fold more than G-1749 did. As the Reviewer correctly pointed out, this difference is not reflected in the cell assay, where the two compounds show comparable activation of the pathway in KMS-11 cells (measured as XBP1s RNA levels). The reason for this phenomenon is unclear. We agree with the Reviewer that this difference may indeed be associated with the difference in the effect that the two activators have on IRE1 oligomerization: G-9807 stabilizes IRE1 dimers, while G-1749 has no effect on dimerization. As a consequence, G-9807 may lead to higher activation levels in biochemical assays than G-1749, while this difference is less obvious in a cell line like KMS-11, where there are IRE1 dimers present at baseline (See Figure 6a).

2. The crystallisation statistics show some strange trends in how the resolution cutoff was selected.

For example the IRE1-0P/G-9807 structure. Here the I/sigI is relatively high in the final shell ~2.5, with a high CC1/2 (0.84). Did the authors try extending the data to higher resolution for some of these structures to see if this improved the electron density?

We thank the Reviewer for bringing this issue to our attention. We have reprocessed the data using smaller resolution shells, and we can see that for the 2.121-2.078 Å shell CC1/2=0.5888 and I/sigI=1.125, which we believe is an acceptable cutoff. If we were to extend the resolution further we would do so to 2.04 Å, for the 2.078-2.039 Å shell reports CC1/2=0.5008 and I/sigI=0.990. However, while a cutoff to CC1/2=0.5 is acceptable, we believe a cutoff of I/sigI=1 would be too low. We are therefore leaving the data deposited in the PDB as is, but have updated Table 1 accordingly using the smaller, more representative, shell size.

Could the authors also include Fo-Fc density maps for their inhibitors as a supplement?

We have added Fo-Fc density omit maps for the small molecules that were co-crystallized with IRE1 in Figure S10.

3. Could the authors highlight the locations of the phosphorylation sites in the Fig 2 structure as this will ease interpretation of later figures.

We have highlighted the phosphates on the kinase activation loop in Figure 2a as sticks and spheres and labeled them. Thank you for the suggestion.

4. It is particularly striking how well the authors have defined activation vs inhibition with the different compounds shown in Fig. 3. This is particularly evident for the comparison of G-1749 vs compound 7, where a single methyl group in the tail completely changes the effect on RNase activity.

The authors highlight this tail region in panel 3d. However this overlay makes it very hard to interpret the hydrogen bonds, and in addition there currently is no figure legend for this panel, making it hard to understand.

We apologize for this oversight and thank the Reviewer for noting the lack of a legend for Figure 3d. We hope that the addition of the legend made things clearer. Additionally, we have changed the color of the hydrogen bonds to gray so that the tail-to-E612 distance measurements can be differentiated from the hydrogen bonds in the picture. We have also moved the distance measurements as a range to the side so that they wouldn't cover one of the key hydrogen bonds.

5. It is very hard to interpret the Fig S8 conserved residue figure. Is there a better way to show this rather than the disjointed surface representation?

We have updated Figure S8 with a new representation where only the C α atoms of the conserved residues are highlighted as spheres, along a cartoon representation of the monomer of IRE1. We hope this is easier to interpret than the previous visualization. We have also updated the legend as necessary.

6. For many of the HDX plots in the supplemental data there are curves that lack some time points. Was this due to overlapping peptides at longer D2O exposures?

One reason for the lack of certain timepoints is that our processing software rejects spectra that don't fit well to mathematical descriptions of what is possible to measure for that peptide (see reference #11 in our updated Supplemental Information), and this can occur for a number of reasons, including the one correctly suggested by the Reviewer. Other reasons could be that the signal nears our cutoff thresholds, and therefore could be variably rejected on these grounds over a time series.

Reviewer #2

The paper by Ferri et al describes the effects of ATP competitive inhibitors on IRE1 activity and suggests that the kinase loop conformation plays an important role in modulating activity. They identify an inhibitor G1749 that displays unusual behaviour which they assess using structural, in vitro and cellular approaches.

Overall, the paper is reasonably interesting and attempts to explain contrasting IRE1 activity on kinase loop conformations. Although this does not explain how this modulates RNase splicing, it does describe an important regulatory aspect of IRE1 activity and therefore warrants publication. It is already known that kinase loop phosphorylation greatly enhances splicing. The technical approach is fairly well thought out and executed well enough but for one major addition/clarification required see below. The conclusion that are drawn are in line with the data presented in the paper.

We thank the Reviewer for the overall supportive remarks regarding our manuscript and approaches. Please see below for responses to the Reviewer's specific questions.

Major point

In figure 6 the data shows that there is an increase in splicing in KMS-11 cells upon addition of G1749. However, one might expect that in these cells G1749 would inhibit splicing since the IRE1 pathway is constitutively activated, suggesting a higher population of phosphorylated IRE1. G1749 was shown convincingly to inhibit splicing of phosphorylated IRE1 in the earlier experiments.

Similarly, in cells that require chemical induction of UPR one would expect that addition of G1749 would activate splicing as there would be low levels of phosphorylated IRE1, particularly before the addition of Tg. This is further supported by the levels of IRE1 dimer when crosslinked. There is a higher proportion of dimers and thus higher splicing in the constitutively activated KMS-11 cell line. Thus, the results obtained using G1749 are the opposite of what one might expect. The key missing information is the direct measure of phosphorylation in these experiments. It is important to assess the levels of IRE1 phosphorylation either by using a specific phospho antibody (p724 antibody) or a phos-tag or similar.

Although KMS-11 cells exhibit constitutive XBP1s, as shown in Figure 6a, these cells do not display constitutive IRE1 phosphorylation. We have included below an experiment where KMS11 cells were treated with Thapsigargin over four hours. Phosphorylated IRE1 (p-IRE1) was detected using an antibody previously published by our laboratory (see T.-K. Chang et al. Mol Cell. 2018, 71(4): 629-636). In KMS11 cells, ER stress by Thapsigargin leads to IRE1 autophosphorylation as early as 30 minutes after the start of treatment, but it then subsides over time. In the experiment reported in Figure 6a, the blot for pIRE1 was an empty blot.

Whereas IRE1 phosphorylation supports RNase activation, it is not absolutely required for this function. Indeed, in MDA-MB-231 cells, CMV-promoter-driven overexpression of the kinase-dead D688N mutant of IRE1, which is incapable of autophosphorylation, drives XBP1 splicing nevertheless (Fig. 6d). Thus, IRE1 can form active dimers even in the absence of phosphorylation, provided that its concentration is sufficiently high—as is the baseline case in KMS11 cells (Fig. 6a, first lane on the left and experiment reported above), or transfected state of MDA-MB-231 cells (Fig. 6d, second and third lane on the left). Accordingly, G-1749 can augment RNase activity of pre-existing IRE1 dimers in KMS11 cells, increasing XBP1 splicing.

As for the same experiment in MDA-MB-231 cells, shown in Figure 6b, the Reviewer is correct in expecting no detectable pIRE1 at baseline: we have added the pIRE1 blot that shows absence of pIRE1 in the negative control and presence of pIRE1 in the positive control after treatment with Tg. As expected, G-1749 and AMG-18 did not have an effect on IRE1 phosphorylation in the absence of Tg, while G-9807 seems to slightly enhance IRE1 phosphorylation, which is likely attributable to an off-target effect due to its relatively low kinase selectivity.

The lack of IRE1 activation by G-1749 in MDA-MB-231 cells is consistent with the hypothesis that G-1749 activates pre-associated (unphosphorylated) IRE1, because this cell line—in contrast to KMS11—has much lower endogenous expression of IRE1 and lacks constitutive dimers. Further supporting this hypothesis, Figure 6d shows that activity of G-1749 in MDA-MB-231 cells was rescued by expression of high levels of kinase-dead IRE1, which favors self-association, but bypasses phosphorylation.

Minor points (not required to do).

In the SV experiments the concentration is not necessarily shown as molar excess of inhibitor, but a 1:1 ratio which is fine but then they should state this.

The ratio is 1:1.15, which we consider slight molar excess. We have added the word “slight” to lines 125, 144, 169, 172, 433. The reason for not using a higher molar excess is to avoid a high background from the compound’s absorbance at 280 nm (our readout during the experiments).

It would be nice to see a secondary measure of monomer-dimer IRE1 bound to inhibitor using cross linking in Figure 1 and throughout.

Both Reviewer #2 and #3 have raised the need for an orthogonal method to SV-AUC to detect recombinant IRE1 dimers and oligomers.

We think SV-AUC is the most relevant method for this study, as it is a direct solution measurement of molecular interactions in which the analysis proceeds from first principles. It is a thermodynamically and hydrodynamically rigorous quantitative method, especially when the recombinant protein is not labeled with fluorescent molecules for detection. Conversely, results of crosslinking reactions vary dramatically depending on the chemical nature of the crosslinking reagent, and are highly dependent on the concentration of these reagents, protein concentration, and length and temperature of the reaction. Crosslinking methods are not quantitative and in fact have the potential to alter the equilibrium between bound and unbound species in solution, making quantitation impractical. Furthermore, the labeling of molecular species is not site specific and the crosslinked species may or may not be physiologically and mechanistically relevant. This may also result from the tendency of crosslinking experiments to stabilize very weak and transient interactions.

We have nonetheless performed crosslinking experiments of unphosphorylated IRE1 in the presence of G-1749 and G-9807.

We utilized DSS, a 6-atom-long crosslinker, the same used in the main text to observe dimers and oligomers in cell lysates (see Figure 6). Three different concentrations of IRE1 (8, 80, and 800 nM) were incubated with large excess of compounds (5 μ M) on ice for 40 minutes. DSS was added to the samples at a final concentration of 250 μ M, followed by incubation for 1 hour on ice. The reaction was quenched using 1 μ L of 1M TRIS pH 7.5 for 15 min on ice. The samples were then run on SDS-PAGE gel at 100V for 3 hours then electro-transferred to nitrocellulose membranes for western blotting with IRE1 antibody to maximize detection. G-9807 stabilized dimers and oligomers of unphosphorylated IRE1 significantly better than G-1749, which, at the lowest concentration of IRE1 (8 nM) does not have any effects on IRE1 dimer levels compared to DMSO. In this experiment, G-1749 seemed to enrich cross-linked IRE1 dimers compared to DMSO control: the reason may be that crosslinking tends to amplify such signal by irreversibly capturing these weak interactions, while SV-AUC should reflect the transient and weakly interacting nature of the dimers more faithfully.

We believe our HDX experiments support the SV-AUC data: G-1749 binding caused destabilization of the kinase regions involved in dimer formation (Figure 4d), unlike G-9807 binding, which stabilized the region. Our cell-based experiments also support the inability of G-1749 to modulate the oligomeric state of IRE1, as the molecule is capable of activating the pathway only in a cell line like KMS-11, where IRE1 is present at high concentrations and can self-associate in the absence of the compound through its luminal domain. Crosslinking of lysates in MDA-MB-231 showed no stabilization of dimers and oligomers in the presence of G-1749, compared to DMSO, while G-9807 increased the concentration of dimers in the lysate.

If the data is available, is the salt-bridge K599-E612 intact in G1749-IRE13P structure?

We do not have a co-crystal structure of G-1749 in complex with IRE1 3P. We have attempted to determine it but we could not obtain high quality crystals for this complex. Instead, we have a co-crystal structure of a very similar compound in the G-1749 SAR, compound 2, in complex with IRE1-3P (Figure 5c). In this co-crystal structure, the K599-E612 salt bridge is not intact, as we report in line 466 of the updated main text.

It would be nice to see a mutation that disrupts the above-mentioned salt bridge and then observe its effects on splicing and dimer formation in solution.

We thank the Reviewer for the suggestion. Mutation K599A (the lysine involved in the aforementioned salt bridge) is routinely utilized in IRE1 research as a well-characterized kinase-dead mutant. We utilize this mutation in the cell experiment reported in Figure 6d (reconstitution of IRE1 in MDA-MB-231 IRE1 KO cells). The compounds were able to bind and modulate this construct as effectively as WT in this experiment. Additionally, the mutant is capable of forming dimers, as evident in the same Figure. We believe that experiments with K599A were extensively performed in IRE1 literature to characterize effects on kinase and RNase activity, and IRE1 association in dimers and oligomers, and as such, redundant with the available information and beyond the scope of our work.

It would be nice to see a mutation that disrupts the b2b dimer based on the structure and then to measure if this then inhibits dimer in solution or not.

We thank the Reviewer for the suggestion. Mutations that disrupt the B2B dimerization interface based on the dimer crystal structure (such as D620A) have been studied before and extensively utilized as RNase-inhibiting mutations, so we believe experiments with these mutations would be overlapping with existing literature and beyond the scope of our work. Of note, we do have recombinant IRE1 D620A and other similar mutants and we have found it difficult to demonstrate that they effectively disrupt IRE dimers.

Do the cellular crosslinking experiments show oligomer formation such as tetramer or larger?

It is difficult to determine the size of the observed crosslinked bands. We sometimes observe multiple bands (as many as three – see Figure 6a and 6b) at the expected molecular weight (MW) for the IRE1 dimer (250 kDa). At least one of the bands should be dimeric full-length IRE1. The two other bands may represent a different conformation of IRE1 at a similar molecular weight, such as the face-to-face conformation, which would be expected to run differently in the gel, or a higher-order oligomer, or IRE1 crosslinked to other protein binding partners. In Figure 6c, right side of the panel, third band from the left (and Figure 6d), the overexpression of IRE1 in reconstitution experiments leads to the appearance of even higher MW bands and even a small amount of IRE1 that remains in the loading well: we believe these may be higher order oligomers.

Reviewer #3

This manuscript presented by Ferri et. al., features the development of IRE1-selective kinase inhibitor-RNase activators. The authors have identified two novel allosteric activators, G-9807 and G-1749, of which G-1749 displays high kinase-selectivity. The mechanism of activation for these novel activators was rationalized with co-crystal structures of IRE1, HD-MX, and both biochemical and cellular RNase activity assays. From these experiments the authors concluded that G-9807, typical of type I kinase inhibitors, favors IRE1 B2B dimer formation and consequently RNase activity. Conversely, G-1749 accesses a region they call the front pocket, which stabilizes IRE1's activation loop in a distinct conformation. As a consequence, G-1749 inhibits phosphorylated IRE1 (IRE1-3P), but activates dephosphorylated IRE1 (IRE1-0P). G-1749 activates IRE1 through an unknown mechanism as no B2B dimers were observed. This work culminates with a cellular experiment, where the authors show that G-1749 is only capable of activating pre-associated IRE1. This study presents a comprehensive analysis of IRE1 kinase inhibitor/RNase activators, incorporating crystallography and HD-MX alongside biochemical and cellular characterization. The molecules presented in this work are a nice addition to the chemical tools that are currently available for IRE1 and the UPR. Specifically, the generation of IRE1-specific allosteric activators are the first of their kind and will be an essential tool to understanding IRE1's unique role as part of the UPR. While the manuscript is well-written, the authors will need to address some concerns below before the manuscript can be accepted for publication.

We thank the Reviewer for the overall positive remarks and constructive suggestions regarding our manuscript and approaches. Please see below for responses to the Reviewer's specific questions.

1. The concentrations required for SV-AUC experimentation are incredibly high. The lowest concentration of IRE1 used in these assays is 3.3 μ M. However, although the concentration of IRE1 used in the RNase assays is not included in the figure legends or experimental (which should also be addressed), it can be assumed that these assays were ran under Michaelis-Menten kinetics where $[IRE1] \ll [RNA]$ used. This would suggest that concentrations of IRE1 used in the RNase assay to obtain robust XBP1 cleavage are much lower than those used in SV-AUC experiments (see Figure S6). How do the authors justify measuring K_{dimer} at such high concentrations, while also using much lower concentrations in their RNase assays (assuming dimerization is necessary for activity in this assay)?

We thank the Reviewer for bringing to our attention that we have not specified reagent concentrations in our RNase assay (we only did in one of the supplementary Figures and we apologize for the oversight). As the reviewer has correctly stated, we have optimized the concentration of IRE1 and RNA substrate to maintain signal linearity during the kinetic assay, especially in the presence of activators. The final conditions were: 8 nM IRE1 LKR and 200 nM RNA. In the case of the graphs reported in Figure 5, IRE1 KR-0P was used at 22 nM with RNA at 200 nM, while IRE1 KR 3P was used at 2

nM with RNA at 500 nM. The reason for changing RNA concentration in the latter conditions is that 200 nM RNA would be extinguished before the end of the experiment (1 hour) when using very active IRE1 3P, even at very low concentration of protein, while KR-0P was less active than IRE1 LKR and as such it required a higher protein concentration in the assay. We have added these specifications to the appropriate Methods section to correct our oversight.

We completely agree with the Reviewer regarding the discrepancy between the IRE1 dimerization constant measured by SV-AUC (ca. 65 μ M) and the protein concentration sufficient for activity in solution (2-20 nM). This is a conundrum that is not new to the field of IRE1 research. There are additional aspects of IRE1 behavior that add to this incongruence: the cytoplasmic constructs of IRE1 (KR and LKR) do not elute as dimers in size exclusion chromatography, even when fully phosphorylated, and even at elevated concentrations (for example during purification). Similarly, when attempting to assess oligomerization state by SEC-MALS, we have never been able to isolate a dimeric peak or observe signs of oligomerization, even when using low-salt buffers. Yet, the evidence for IRE1 cleaving its substrates as a dimer or higher-order oligomer is well-established and compelling in the literature. Our explanation for these incongruences is that the cytoplasmic domains of IRE1 may associate in transient dimers. This results in very low concentration of dimers in solution, difficult to measure by available biophysical techniques, but perhaps sufficient for RNA cleavage. Transient dimers in solution would also result in a high dimerization constant measured by SV-AUC. This explanation makes biological sense, as the association of IRE1 in a cellular context is regulated by the luminal domain, which, unlike the cytoplasmic region of the protein, has a very high propensity to association during purification and in other biophysical experiments.

Of note, the concentration of IRE1 we used in our SV-AUC experiments (3.3 μ M, 10 μ M, 30 μ M) is in the same range as published SV-AUC experiments with IRE1: Korennykh et al. (Nature, 2009) used 13.5 μ M yeast IRE1, and Sanches et al. (Nature Communications, 2014) used 18 μ M human IRE1.

2. A major claim of the paper is that G-1749 does not stabilize dimers, contrary to previously published work on allosteric activators. This would fundamentally change how we think about the kinase-RNase allosteric network. Considering this, more needs to be done to convince the reader that G-1749 does not stabilize dimers. This claim could be bolstered by performing additional chemical crosslinking experiments or FRET-based dimerization experiments. Alternatively, the argument that G-1749 does not stabilize dimerization would be much more convincing if the authors displayed activation of IRE1 with G-1749 using IRE1-0P with a mutant in the B2B dimer interface (e.g. D620). In general, I think the authors current evidence claiming G-1749 functions to activate IRE1 independently of dimerization needs stronger biochemical evidence.

We thank the Reviewer for the comments, as they highlight that perhaps we were not entirely clear in explaining what we think may be the mechanism of G-1749. As mentioned in our response to the previous point, we do subscribe to the literature evidence that IRE1 must cleave RNA minimally as a dimer. G-1749 does not seem to substantially stabilize IRE1 dimers in our experiments (see Figure 1 and crosslinking

experiments included in response to Reviewer #2 above), but we do not believe that G-1749 activates a monomeric form of IRE1. On the contrary, we believe that the compound “boosts” the activity of dimeric IRE1 (including the transient ones we believe may be present in solution but not easily measurable – see our response to point #1) and is unable to activate monomeric IRE1 (based on the cellular experiments in Figure 6, see lines 593-595 and 598-599 in the updated main text. See also in the Discussion section lines 647-654). We think G-1749 binds to both monomers and pre-existing IRE1 dimers (and oligomers), but only boosts the activity of the dimers (and perhaps oligomers), while being agnostic to monomeric IRE1. Unlike “traditional” allosteric activators, which skew the IRE1 monomer-dimer equilibrium towards dimers (see lines 174-176), G-1749 does not seem to significantly alter the dimerization constant. For these reasons, testing the activity of G-1749 on a dimerization-deficient mutant of IRE1 is not an experiment that is pertinent to our mechanistic hypothesis: we do not expect G-1749 to activate the D620A mutant or other mutants that would impair IRE1 association.

We have revised the manuscript to better explain our hypothesis and we hope that it is clearer following these revisions:

- Lines 626-628: added: “The evidence for IRE1 cleaving its substrates as a dimer or higher-order oligomer is well-established and compelling in the literature.^[Refs] We do not think that G-1749 activates a monomeric form of IRE1: [cont’d]”
- line 698, added “in a B2B dimer”

3. The SAR analysis of front pocket binding is puzzling. The authors state the front pocket has a limited tolerance for chemical substituents; stating that bulkier substituents cannot occupy the front pocket. But inhibitors that have smaller substituents (comp 4 and 5), that presumably could fit into the front pocket are shown to inhibit the RNase domain. Considering that all allosteric RNase inhibitors tested in this focused SAR lack groups that can accept electrons from the carboxylate of Glu612 (such as a carbamate or acrylamide), couldn't this instead suggest that to be an activator, there needs to be an electron accepting group to coordinate Glu612 of the α C-helix anchoring it an inward position?

We do not consider Inhibitors 4 and 5 “smaller” than parent compound G-1749 in the way we use this qualifier in the SAR description: compounds 4 and 5’s “tail” is shorter in length, but the substitution of the carbamate oxygen of G-1749 for a methylene in compounds 4 and 5 enlarges the tail’s Van der Waals volume, instead of decreasing it, and makes it therefore more likely to insult E612 and not fit into the front pocket. Additionally, there were carbamates in our inhibitor series: compounds 7, 8, 9, 11 are carbamates. Finally, in the crystal structure of G-1749 (and G-7658), the compound’s tail establishes a direct hydrogen bond with E612 via the NH group on the tail (see Figure 2c and revised Figure 3d): all compounds from the SAR bear the same NH moiety, in the form of either an amide or carbamate. Therefore, we don’t believe a direct coordination with E612 to be the discriminant for activation versus inhibition.

We think this misunderstanding may have stemmed from a typo we found in the text at line 298 in the updated main text, where we mistakenly stated that the “nitrogen” of the G-1749 carbamate tail was substituted by a carbon in compounds 4 and 5, while instead it’s the oxygen that was substituted by a methylene group. We apologize for this error and have corrected the text accordingly. We hope this change makes the SAR interpretation clearer.

4. Table S3 should include SEM values.

Table S3 reports the calculated EC₅₀ values for the curves reported in Figure 1 and 2 for the purpose of direct comparison of all compounds. We have added SD values to the table for those curves.

5. In Figure 6 the authors use DSS, an amine-reactive chemical crosslinker, to chemically crosslink IRE1. This is problematic as AMG-18 and G-1749 both have free-amines that can react with DSS. Considering the stoichiometric excess of small molecule to endogenous IRE1 wouldn't one expect a majority of the DSS to be quenched by inhibitor?

The highest concentration of compounds in the experiments was 1 μ M, but the actual concentration of compound that penetrates the cells is expected to be significantly lower: this is relevant, as the crosslinking happened after eliminating cell media containing compound that did not penetrate the cells, washing cells with PBS, and then lysing them. Because of that, we do not expect significant contribution of compound reacting with DSS. Nonetheless, we have performed a proof-of-concept experiment where recombinant IRE1 KR fully phosphorylated (3P) at 800 nM in HEPES pH 7.5, 150 mM NaCl buffer was incubated with or without piperidine, the amine-containing chemical fragment present in these inhibitors, at a final concentration of 5 μ M (in DMSO) and then crosslinked with 250 nM DSS for one hour at room temperature. Crosslinked samples were quenched with 50 mM TRIS pH 7.5 for 15 minutes at room temperature. This experiment mimics the conditions of lysate crosslinking in our cell experiments and the reactivity of the piperidyl group of AMG-18 and G-1749, but it eliminates the modulation of IRE1 oligomerization by the compounds.

The experiment showed that IRE1 dimerization and oligomerization in the presence of DSS was not impaired by large excess of piperidine, which indicates that the piperidinyl group of the compounds does not significantly quench DSS.

6. *Authors need to show that G-1749 and G-9807 are engaging IRE1 to a similar extent in cells. The results shown could be due to different levels of engagement in the respective cells lines. Additionally, because phosphorylation state is important to the mechanism of G-1749, the authors should also characterize the phosphorylation status of IRE1 in cells using phostag gels.*

With respect to levels of phosphorylated IRE1 in Figure 6, please see our response to the second point made by Reviewer #2.

We interpret that the Reviewer means “IRE1 binding” by the word “engagement”. If so, then this raises a valid point, which we indeed considered while designing the experiments reported in Figure 6: perhaps the different activity of G-1749 in KMS-11 versus MDA-MB-231 cells stems from differences in permeability or other factors across cell lines. That is the reason why we have established an IRE1 KO cell line in Figure 6d using MDA-MB-231 cells: observing a rescue of G-1749 activity when transfecting IRE1 at higher concentrations than WT in these cells gave us confidence that the absence of IRE1 modulation by G-1749 in MDA-MB-231 cells was not attributable to lower engagement compared to KMS-11.

Nevertheless, we performed an additional experiment where we treated MDA-MB-231 cells with Thapsigargin (Tg) for two hours, and then added our compounds to the cells and incubated them for four more hours: our rationale was that if G-1749 were to engage IRE1 successfully then we would observe inhibition of IRE1 autophosphorylation due to IRE1 kinase binding. This experiment was run side-by-side with treatment of the cells in absence of Tg, which replicated what we reported in Figure 6b: G-1749 did not appreciably increase XBP1 splicing at a compound concentration of 1 μ M, while G-9807 did. In the presence of Tg, we could detect pIRE1 at baseline and we observed complete inhibition of pIRE1 by both AMG-18 and G-1749, indicating that G-1749 successfully binds IRE1 in the MDA-MB-231 cell line. Interestingly, G-1749 mostly inhibited XBP1 splicing in this context, perhaps pointing to a difference in kinetics between pIRE1 inhibition (G-1749 would inhibit XBP1 splicing when IRE1 is phosphorylated) and XBP1 splicing activation (which would happen only once IRE1 is not phosphorylated). G-9807 unexpectedly did not inhibit autophosphorylation, a

phenomenon that we attribute to the compound's promiscuity that may cause indirect ER stress, balancing out the levels of IRE1 phosphorylation.

We have verified that G-9807 indeed does inhibit autophosphorylation (as one would expect based on competitive binding with ATP): incubation of recombinant unphosphorylated IRE1 with large excess of either G-9807 or G-1749 significantly impaired autophosphorylation in the presence of ATP and $MgCl_2$ compared to DMSO.

Generally, these results show how difficult it is to interpret effects in cells of small molecule kinase inhibitors that are RNase activators, and especially when the compound inhibits numerous other kinases such as G-9807.

7. Can the authors speculate further on why G-1749 would only activate pre-associated IRE1? Isn't this contrary to what is observed in vitro, where G-1749 can activate IRE1-0P using concentrations of IRE1-0P that are well below 65 μ M (the reported KDIMER for IRE1-0P)?

This point is related to the first point raised by the Reviewer and it is a very valid one. As explained in our response to the Reviewer's first point, we think that IRE1 cleaves RNA as a dimer, as extensively demonstrated in the literature. As such, we speculate that there has to be a certain amount of active dimer, albeit transient and not measurable at these concentrations by the methods we used, at 2-22 nM concentration when running our RNA cleavage assay. This amount of dimer is the amount that G-1749 activates. The hypothesis that G-1749 activates pre-associated IRE1 was formulated based on our SV-AUC data reported in Figure 1 showing lack of dimer stabilization and the cell experiments reported in Figure 6, where we observed G-1749 activating IRE1 when it was present in cells in high concentration (KMS-11 and kinase-dead IRE1 overexpression in MDA-MB-231 IRE1 KO), otherwise being relatively inert in cell lines that did not present IRE1 pre-association in dimers and oligomers (MDA-MB-231 WT).

REVIEWERS' COMMENTS

Reviewer #1 (Remarks to the Author):

The authors have done an excellent job in answering my responses, I think this will make an excellent contribution to the field. I have only one very small final correction:

Minor

In Fig 3b the compounds are graphed without a clear description of which symbol represents which compound. It can be inferred based on the alignment of each compound with each curve, but this should be included (see Fig 3c as a better way of demonstrating this, as it clearly shows what compound is represented by which symbol).

Reviewer #3 (Remarks to the Author):

The authors have satisfactorily addressed this reviewer's points in the revised manuscript. With these additional edits, I'm please to recommend this manuscript for publication.

Response to Reviewers' comments for manuscript NCOMMS-20-07191

(2nd round of review)

Changes to Figures in the Manuscript

Figure 3

- Panel b, we added symbols on the side to indicate which symbol represent which compound

Responses to reviewers

(reviewers' comments are quoted in blue and italicized)

Reviewer #1

The authors have done an excellent job in answering my responses, I think this will make an excellent contribution to the field. I have only one very small final correction:

Minor

In Fig 3b the compounds are graphed without a clear description of which symbol represents which compound. It can be inferred based on the alignment of each compound with each curve, but this should be included (see Fig 3c as a better way of demonstrating this, as it clearly shows what compound is represented by which symbol).

We sincerely thank the reviewers for the positive remarks.

We have addressed the request by the reviewer and added symbols on the side of Fig 3b to clarify which symbol represents which compound.

Reviewer #3

The authors have satisfactorily addressed this reviewer's points in the revised manuscript. With these additional edits, I'm please to recommend this manuscript for publication.

We sincerely thank the reviewer for the recommendation.